# Deep Backtracking Counterfactuals for Causally Compliant Explanations

## Abstract

Counterfactuals can offer valuable insights by answering what would have been observed under altered circumstances, conditional on a factual observation. Whereas the classical interventional interpretation of counterfactuals has been studied extensively, *backtracking* constitutes a less studied alternative where all causal laws are kept intact. In the present work, we introduce a practical method for computing backtracking counterfactuals in structural causal models that consist of deep generative components. To this end, we impose conditions on the structural assignments that enable the generation of counterfactuals by solving a tractable constrained optimization problem in the structured latent space of a causal model. Our formulation also facilitates a comparison with methods in the field of counterfactual explanations. Compared to these, our method represents a versatile, modular and causally compliant alternative. We demonstrate these properties experimentally on a modified version of MNIST and CelebA.

## 1 Introduction

In recent years, there has been a surge in the use of deep learning for causal modelling (Pawlowski et al., 2020; Kocaoglu et al., 2018; Goudet et al., 2018). The integration of deep learning in causal modelling combines the potential to effectively operate on high-dimensional distributions, a strength inherent to deep generative modeling, with the capability to answer inquiries of a causal nature, thus going beyond statistical associations. At the apex of such inquiries lies the ability to generate scenarios of a counterfactual nature—altered worlds where variables differ from their factual realizations, hence aptly termed *counter to fact* (Pearl, 2009; Bareinboim et al., 2022). Counterfactuals are deeply ingrained in human reasoning (Roese, 1997), as evident from phrases such as *"Had it rained, the grass would be greener now"* or *"Had I invested in bitcoin, I would have become rich"*.

Constructing counterfactuals necessitates two fundamental components: (i) a sufficiently accurate world model with mechanistic semantics, such as a structural causal model; and (ii) a sound procedure for deriving the distribution of all variables that are not subject to explicit alteration. The latter component has been a subject of debate: While the classical literature in causality constructs counterfactuals by actively manipulating causal relationships (*interventional counterfactuals*), this approach has been contested by some psychologists and philosophers (Rips, 2010; Gerstenberg et al., 2013; Lucas & Kemp, 2015). Instead, they have proposed an account of counterfactuals where alternate worlds are derived by tracing changes back to background conditions while leaving all causal mechanisms intact. This type of counterfactual is therefore termed *backtracking counterfactual* (Jackson, 1977). Due to the preservation of causal mechanisms, backtracking counterfactuals allow for gaining faithful insights into the structural relationships of the data generating process, which render them a promising opportunity in practical domains such as medical imaging (Sudlow et al., 2015), biology (Yang et al., 2021a) and robotics (Ahmed et al., 2021). Recently, von Kügelgen et al. (2023) have formalized the backtracking counterfactual within the structural causal model framework. However, implementing this formalization for deep structural causal models poses challenges due to multiple computationally intractable steps, such as marginalizations and the evaluation of distributions that are computationally intractable. The present work addresses these challenges and offers a computationally tractable implementation by framing the generation of counterfactuals as a constrained optimization problem. The optimization is solved with an iterative algorithm, which linearizes the reduced form of the structural causal model. These measures provide effective remedies for generating counterfactual scenarios in multi-variable data with known causal relationships.

Figure 1: **Visualization of DeepBC for Morpho-MNIST.** We generate a counterfactual (green) image img$^*$ and thickness $t^*$ with antecedent intensity $i^*$ for the factual, observable realizations (blue) img, $t$, $i$. Our approach finds new latent variables $\mathbf{u}^*$ that minimize distances $d_i$ to the factual latents $\mathbf{u}$, subject to rendering the antecedent $i^*$ true. The causal mechanisms in the factual world remain unaltered in the counterfactual world. In this specific distribution, thickness and intensity are positively related, thus rendering the image both more intense and thicker in the counterfactual. Dependence of $f_i$ on graphical parents is omitted for simplifying visual appearance.

Furthermore, the present work serves as a bridge between causal modelling and practical methods in the field of high-dimensional counterfactual explanations, which, despite its similar nomenclature, has evolved largely independently from the field of counterfactuals in causality.

We summarize our main contributions as follows:

- We introduce a computationally tractable method called *deep backtracking counterfactuals* (DeepBC) for computing backtracking counterfactuals in deep structural causal models (§ 3). Our method exhibits multiple favorable properties such as versatility, causal compliance and modularity (§ 3.2).

- We show the relation between our method and the field of counterfactual explanations and elucidate how our method can be understood as a general form of the popular method proposed by Wachter et al. (2017) (§ 3.1).

- We demonstrate the applicability and distinct advantages of our method through experiments on two data sets, in comparison to existing methods. Specifically, we apply our method to Morpho-MNIST and the CelebA data set (§ 4).

**Overview.** Section § 2 introduces structural causal models (§ 2.1), the deep generative models that are employed subsequently (§ 2.2), interventional and backtracking counterfactuals (§ 2.3) and counterfactual explanations (§ 2.4). In Section § 3, we propose our method called *deep backtracking counterfactuals* (DeepBC) and discuss its relation to methods in the field of counterfactual explanations (§ 3.1) and its implementation (§ 3.3). In Section § 4, we perform experiments on Morpho-MNIST (§ 4.1) and CelebA (§ 4.2) that highlight the versatility, modularity and causal compliance of our method. In Section § 5, we present a compact related work. A more comprehensive related work section is included in App. E. We then discuss the limitations of our work in § 6 and conclude with a short summary in § 7.

The following section introduces structural causal models and backtracking counterfactuals, which sets the stage for introducing our method in § 3.

## 2 SETTING & PRELIMINARIES

**Notation.** Upper case $X$ denotes a scalar or multivariate continuous random variable, and lower case $x$ a realization thereof. Bold $\mathbf{X}$ denotes a collection of such random variables with realizations $\mathbf{x}$. The components of $\mathbf{x}$ will be denoted by $x_i$. We denote the probability density of $X$ by $p(x)$.

## 2.1 Structural Causal Models

Let $\mathbf{X} = (X_1, X_2, ..., X_n)$ be a collection of potentially high-dimensional observable "endogenous" random variables. For instance, $X_1$ could be a high-dimensional object such as an image and $X_2$ a scalar feature variable. The causal relationships among the $X_i$ are specified by a directed acyclic graph $G$ that is known. A structural causal model is characterized by a collection of structural equations $X_i \leftarrow f_i(\mathbf{X}_{\text{pa}(i)}, U_i)$, for $i = 1, 2, ..., n$, where $\mathbf{X}_{\text{pa}(i)}$ are the causal parents of $X_i$ as specified by $G$ and $\mathbf{U} = (U_1, U_2, ..., U_n)$ are "exogenous" latent variables. The acyclicity of $G$ ensures that for all $i$, we can recursively solve for $X_i$ to obtain a deterministic expression in terms of $\mathbf{U}$. Thus, there exists a unique function that maps $\mathbf{U}$ to $\mathbf{X}$, which we denote by $\mathbf{F}$:

$$\mathbf{X} = \mathbf{F}(\mathbf{U}), \tag{1}$$

and is known as the reduced-form expression. Hence, $\mathbf{F}$ induces a distribution over observables $\mathbf{X}$, for any given distribution over the latents $\mathbf{U}$. For the remainder of this work, we assume causal sufficiency (Spirtes, 2010) (no unobserved confounders), which implies joint independence of the components of $\mathbf{U}$.

## 2.2 Deep Invertible Structural Causal Models

In this work, we make the simplifying assumption that $f_i(\mathbf{x}_{\text{pa}(i)}, \cdot)$ is invertible for any fixed $\mathbf{x}_{\text{pa}(i)}$[1], such that we can write

$$U_i = f_i^{-1}(\mathbf{X}_{\text{pa}(i)}, X_i), \quad i = 1, 2, ..., n.$$

Under this assumption, the inverse $\mathbf{F}^{-1}$ of the mapping in (1) is guaranteed to exist, and we can write $\mathbf{U} = \mathbf{F}^{-1}(\mathbf{X})$. We assume that all $f_i$ are given as (conditional) deep generative models[2], trained separately for each structural assignment (Pawlowski et al., 2020). We consider the following two classes of models, both of which operate on latent variables with a standard Gaussian prior.

**Conditional normalizing flows** (Rezende & Mohamed, 2015; Winkler et al., 2019) are constructed as a composition of invertible functions, hence rendering the entire function $f_i$ invertible in $u_i$. In addition, they are chosen such that the determinant of the Jacobian can be compted efficiently. These two attributes facilitate efficient training of $f_i$ via maximum likelihood.

**Conditional variational auto-encoders** (Kingma & Welling, 2014; Sohn et al., 2015) consist of separate encoder $e_i$ and decoder $d_i$ networks. These modules parameterize the mean of their respective conditional distributions, i.e., $U_i | \mathbf{x}_{\text{pa}(i)}, x_i \sim \mathcal{N}(e_i(\mathbf{x}_{\text{pa}(i)}, x_i), \text{diag}(\boldsymbol{\sigma}_e^2))$ and $X_i | \mathbf{x}_{\text{pa}(i)}, u_i \sim \mathcal{N}(d_i(\mathbf{x}_{\text{pa}(i)}, u_i), \mathbf{I}\sigma_d^2)$. Through joint training of $e_i$, $d_i$ and variance vector $\boldsymbol{\sigma}_e^2$ using variational inference, $e_i$ and $d_i$ become interconnected. Theoretical insights by Reizinger et al. (2022) support the use of an approximation, where the decoder effectively inverts the encoder, that is,

$$x_i = f_i(\mathbf{x}_{\text{pa}(i)}, f_i^{-1}(\mathbf{x}_{\text{pa}(i)}, x_i)) \approx d_i(\mathbf{x}_{\text{pa}(i)}, e_i(\mathbf{x}_{\text{pa}(i)}, x_i)).$$

## 2.3 Interventional and Backtracking Counterfactuals

Given a factual observation $\mathbf{x}$ and a so-called antecedent $\mathbf{x}_S^* = (x_i^* : i \in S)$ for a given subset $S \subset \{1, 2, ...., n\}$, we define a counterfactual as some $\mathbf{x}^* = (x_1^*, x_2^*, ..., x_n^*)$ consistent with $\mathbf{x}_S^*$. We view $\mathbf{x}^*$ as an answer to the verbal query *"What values ($\mathbf{x}^*$) had $\mathbf{X}$ taken instead of the given (observed) $\mathbf{x}$, had $\mathbf{X}_S$ taken the values $\mathbf{x}_S^*$ rather than $\mathbf{x}_S$?"*. In the present work, we consider interventional and backtracking counterfactuals. Both generate distributions over counterfactuals whose random variables we refer to as $\mathbf{X}^*$. We only provide a conceptual notion and refer the reader to App. A.1 for a more rigorous formalism for both types of counterfactuals.

**Interventional counterfactuals** render the antecedent true via modification of the structural assignments $(f_1, f_2, ..., f_n)$, which leads to a new collection of assignments $(f_1^*, f_2^*, ..., f_n^*)$. Specifically, these new structural assignments are constructed such that the causal dependence on the causal parents of all antecedent variables $\mathbf{X}_S^*$ is removed: $f_i^* = x_i^*$ for $i \in S$ and $f_i^* = f_i$ otherwise. Such a modification can be understood as a *hard intervention* on the underlying structural relations.

---

[1]also known as *bijective generation mechanism* (e.g., see Nasr-Esfahany et al. (2023))

[2]it was shown by Javaloy et al. (2023) that we can identify the true underlying structural equations from observational data, if we furthermore assume that all $f_i$ are diffeomorphic and $u_i$ are univariate and real-valued.

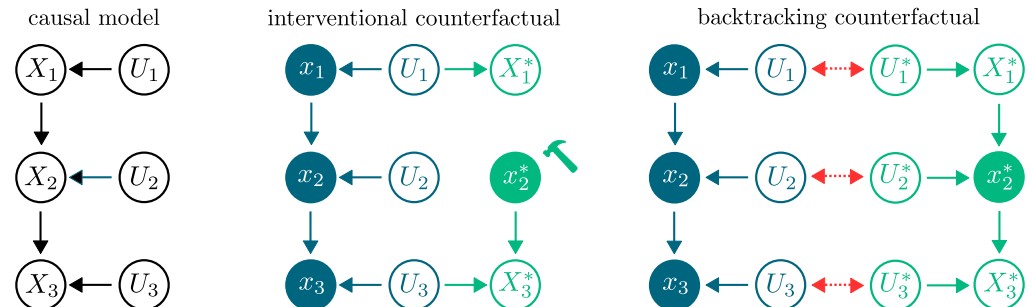

Figure 2: **Difference between interventional and backtracking counterfactuals on an example.** Variables that are conditioned on correspond to filled circles. Interventional counterfactuals perform a hard intervention (indicated by a hammer) $X_2^* \leftarrow x_2^*$ with antecedent $x_2^*$ (i.e., $S = \{2\}$) in the counterfactual world (green). Backtracking counterfactuals, on the contrary, construct this counterfactual world via introducing a new set of latent variables $\mathbf{U}^*$ that depend on $\mathbf{U}$ via a backtracking conditional (red).

**Backtracking counterfactuals** leave all structural assignments unchanged. In order to set the antecedent $\mathbf{x}_S^* \neq \mathbf{x}_S$ true, they trace differences to the factual realization back to (ideally small) changes in the latent variables $\mathbf{U}$. These modified latent variables are represented by a new collection of variables $\mathbf{U}^*$ that depend on $\mathbf{U}$ via a backtracking conditional $p(\mathbf{u}^*|\mathbf{u})$ (von Kügelgen et al., 2023), which represents a probability density for computing similarity between $\mathbf{u}$ and $\mathbf{u}^*$ and which we assume to be decomposable, or factorized: $p(\mathbf{u}^* \,|\, \mathbf{u}) = \prod_{i=1}^n p(u_i^* \,|\, u_i)$. By marginalizing over $\mathbf{U}^*$, we obtain the distribution of $\mathbf{X}^* \,|\, \mathbf{x}_S^*, \mathbf{x}$.

Section § 2 concludes by introducing so-called counterfactual explanations. This allows us to compare our method against this formulation in Section § 3.1.

### 2.4 COUNTERFACTUAL EXPLANATIONS

A wealth of prior work in machine learning is concerned about explaining the prediction $\hat{y}$ of a classifier $f_{\hat{Y}}$ with $\hat{y} \leftarrow f_{\hat{Y}}(x)$ through the generation of a new example $x^*$ which is close to $x$, yet predicted as $y^*$, where $y^*$ is a label that differs from the (factual) prediction $\hat{y}$ [3]. The intuitive idea is that contrasting $x^*$ with $x$ yields an interpretable answer as to why $x$ is classified as $\hat{y}$ rather than $y^*$. Formally (see Wachter et al. (2017)), $x^*$ can be obtained as the solution of

$$\arg \min_{x'} d_o\left(x', \; x\right) \quad \text{subject to} \quad f_{\hat{Y}}(x') \; = \; y^*, \tag{2}$$

where $d_o$ represents a distance function between observed variables.

## 3 DEEP BACKTRACKING COUNTERFACTUALS (DEEPBC)

In this work, we propose to generate a counterfactual $\mathbf{x}^*$ for the factual realization $\mathbf{x}$ as a solution to the following constrained optimization problem:

$$\arg \min_{\mathbf{x}'} \sum_{i=1}^n d_i\left(\mathbf{F}_i^{-1}(\mathbf{x}'), \; \mathbf{F}_i^{-1}(\mathbf{x})\right) \qquad \text{subject to} \qquad \mathbf{x}'_S \; = \; \mathbf{x}_S^*, \tag{3}$$

where $d_i$ denotes a differentiable distance function. Intuitively, we can understand this optimization as finding a solution $\mathbf{x}^*$ that is *close* to the factual realization $\mathbf{x}$ in terms of its latent components, while fulfilling the constraint that $\mathbf{x}^*$ is compliant both with the antecedent $\mathbf{x}_S^*$ and with the causal laws. This situation is visualized on the Morpho-MNIST example in Fig. 1. We further note that (3) is equivalent to an optimization problem within the latent space, i.e.:

$$\arg \min_{\mathbf{u}'} \sum_{i=1}^n d_i\left(u_i', \; u_i\right) \quad \text{subject to} \quad \mathbf{F}_S(\mathbf{u}') \; = \; \mathbf{x}_S^*, \; \mathbf{F}(\mathbf{u}) = \mathbf{x}. \tag{4}$$

We obtain the solution of (3) by inserting the solution of (4) into $\mathbf{F}$. In App. A.2, we provide a derivation of DeepBC from the formalization given by von Kügelgen et al. (2023) .

---

[3]We stress that $\hat{Y}$ is the prediction of a model and thus an effect of $X$. In general, $\hat{Y}$ does not agree with $Y$ (the true variable that is not predicted), since $Y$ might not be the cause of $X$ or might be confounded with $X$.

### 3.1 RELATION TO COUNTERFACTUAL EXPLANATIONS

We can recover counterfactual explanations § 2.4 as a special form of DeepBC. To this end, we assume access to two variables with the following structural equations

$$X \leftarrow f_X(U_X) \quad \text{and} \quad \hat{Y} \leftarrow f_{\hat{Y}}(X), \tag{5}$$

where we note that $\hat{Y}$ is not subject to additional randomness $U_{\hat{Y}}$. In this specific case, we observe that the DeepBC optimization problem (3) reduces to

$$\arg \min_{x'} d_X \left( f_X^{-1}(x'), \ f_X^{-1}(x) \right) \quad \text{subject to} \quad f_{\hat{Y}}(x') \ = \ y^*, \tag{6}$$

which can be interpreted as an instance of (2), where distance is measured in an unstructured latent space. From this viewpoint, we can interpret DeepBC as a general form of counterfactual explanations (2) in two ways: Firstly, it accommodates non-deterministic relations among variables, taking into account the influence of noise on all variables. In the aforementioned instance (5), this can be modeled by $Y \leftarrow f_Y(X, U_Y)$. Secondly, DeepBC can account for multiple variables with complex causal relationships. [4]

### 3.2 METHODOLOGICAL CONTRIBUTIONS

We highlight the main contributions of our work in the context of counterfactual explanations, which we demonstrate experimentally in § 4 and App. D:

1. **Versatility.** DeepBC naturally supports complex causal relationship between multiple variables that are potentially high dimensional (e.g., images or scalar attributes), which goes beyond the instance-label setup (5) presented in § 3.1, and supports flexible choices of antecedent variables. Further, it allows for varying the distance functions $d_i$ in (3) to obtain counterfactuals with different properties (see § 3.3), such as variable preservation (see App. D.1) or sparsity (see § 4.2).

2. **Causal Compliance.** A plethora of work has discussed the right choice of distance function between data points for generating counterfactual explanations (see, e.g., Guidotti, 2022). In this context, DeepBC offers a causally compliant solution: Rather than defining similarity directly between observable variables that can lead to violations of causal laws, DeepBC delineates similarity in terms of latent variables, embedded into a causal model. This implies that generated counterfactual explanations are guaranteed to preserve causal relationships since the counterfactual variables are always subject to the causal laws of the factual world.

3. **Modularity.** Structural relations between variables $(f_1, f_2, ..., f_n)$ exhibit disparities across distinct domains. It has been postulated that these disparities tend to manifest sparsely, signifying that many modules $f_i$ demonstrate analogous behavior across different domains (Schölkopf et al., 2021; Perry et al., 2022). Leveraging the explicit incorporation of structural equations, DeepBC offers adaptability to new domains through the straightforward substitution of individual components $f_i$, without the need for relearning the remaining modules. This contrasts with counterfactual explanation methods, which do not incorporate such replaceable modules and thus require relearning of the entire model to handle a domain shift.

### 3.3 ALGORITHMS

We rely on a penalty formulation to approximate (4), leading to an unconstrained optimization problem. Specifically, we aim at minimizing the following objective function with respect to $\mathbf{u}'$:

$$\mathcal{L}(\mathbf{u}'; \mathbf{u}, \mathbf{x}_S^*) \ := \ \sum_{i=1}^{n} d_i(u_i', \ u_i) \ + \ \lambda \left\| \mathbf{F}_S(\mathbf{u}') - \mathbf{x}_S^* \right\|_2^2, \tag{7}$$

where $\lambda > 0$ is a sufficiently large penalty parameter and $\mathbf{u} = \mathbf{F}(\mathbf{x})$.

**DeepBC via Constraint Linearization.** Rather than minimizing (7) via gradient descent, we empirically observe that employing the first-order Taylor approximation of $\mathbf{F}_S$ at $\bar{\mathbf{u}}$ is beneficial, when minimizing the distance $d_i(u_i', \bar{u}_i) \ = \ w_i \cdot \left\| u_i' - \bar{u}_i \right\|_2^2$ with $w_i \ > \ 0$[5], i.e., $\mathbf{F}_S(\mathbf{u}') \approx$

---

[4]For example, there could be a third variable $Z$ related to $X$ and $Y$ in (6) that could be modeled as well.

[5]By default, we set $w_i = 1$, for all $i$. A different choice of weights may be useful in settings where certain variables should be more preserved due to application-specific insights.

$\mathbf{F}_S(\bar{\mathbf{u}}) + \mathbf{J}_S(\bar{\mathbf{u}})(\mathbf{u}' - \bar{\mathbf{u}})$, where $\mathbf{J}_S(\bar{\mathbf{u}}) := \nabla_{\mathbf{u}}\mathbf{F}_S(\bar{\mathbf{u}})^\top$ denotes the Jacobian matrix. As a result of this approximation, (7) is a convex quadratic function in $\mathbf{u}'$ and can therefore be solved for its minimum $\hat{\mathbf{u}}^*$ in closed form:

$$\hat{\mathbf{u}}^* = (\mathbf{W} + \lambda\mathbf{J}_S^\top(\bar{\mathbf{u}})\mathbf{J}_S(\bar{\mathbf{u}}))^{-1}(\mathbf{W}\mathbf{u} + \lambda\mathbf{J}_S^\top(\bar{\mathbf{u}})\tilde{\mathbf{x}}_S^*), \tag{8}$$

where $\tilde{\mathbf{x}}_S^* := \mathbf{x}_S^* + \mathbf{J}_S(\bar{\mathbf{u}})\bar{\mathbf{u}} - \mathbf{F}_S(\bar{\mathbf{u}})$ and $\mathbf{W} := \mathrm{diag}(w_i)$ is a diagonal matrix containing the distance weights $w_i$. A detailed derivation of (8) is provided in App. A.3.

Solving (8) once, starting from the initial condition $\bar{\mathbf{u}} = \mathbf{u}$, does not accurately fulfill the constraint due to the constraint linearization, except for special cases. We thus apply an iterative algorithm similar to the Levenberg-Marquardt method (e.g., Moré (2006)), based on (8) that is specified in Alg. 1. Empirically, we observe Alg. 1 to converge much faster than gradient descent (see App. B.1 for more implementation details and experiments).

---

**Algorithm 1** DeepBC via Constraint Linearization

$\mathbf{u}'_0 \leftarrow \mathbf{u}$
**for** $t = 1, 2, ..., \#\text{it}$ **do**
$\quad \bar{\mathbf{J}}_S \leftarrow \mathbf{J}_S(\mathbf{u}'_{t-1})$
$\quad \tilde{\mathbf{x}}_S^* \leftarrow \mathbf{x}_S^* + \bar{\mathbf{J}}_S\mathbf{u}'_{t-1} - \mathbf{F}_S(\mathbf{u}'_{t-1})$
$\quad \mathbf{u}'_t \leftarrow (\mathbf{W} + \lambda\bar{\mathbf{J}}_S^\top\bar{\mathbf{J}}_S)^{-1}(\mathbf{W}\mathbf{u} + \lambda\bar{\mathbf{J}}_S^\top\tilde{\mathbf{x}}_S^*)$
**end for**

---

**Sparse DeepBC.** We further employ a variant of DeepBC that encourages sparse solutions, where sparsity is measured in $\mathbf{u}$ rather than $\mathbf{x}$. Specifically, we use sparse DeepBC to obtain solutions where only few elements in $\mathbf{u}^*$ differ from $\mathbf{u}$, i.e., $d_i(u'_i, u_i) = \|u'_i - u_i\|_0$, for all $i$, where $\|\cdot\|_0$ denotes the number of nonzero elements. We apply a greedy approach similar to Mothilal et al. (2020), where we start by fixing an integer $M > 0$ for which we desire that $\|\mathbf{u}' - \mathbf{u}\|_0 \leq M$. We then apply an optimization twice: In a first step, we solve for $\mathbf{u}^*$ using DeepBC. Then, we use the $M$ elements of the solution vector with largest $\|u_i - u_i^*\|_2$ and apply DeepBC again only on these elements, while fixing the others to $u_i$. We note that all $u_i$ have a standard Gaussian distribution, so we do not need to weigh $\|u_i - u_i^*\|_2$ by standard deviation/mean absolute distance, see § 2.2.

## 4 EXPERIMENTS

We run experiments as to contrast DeepBC to existing ideas and showcase its properties and abilities as outlined in § 3.1. We provide all technical details about the implementation in App. B.

### 4.1 MORPHO-MNIST

**Experimental Setup.** We use Morpho-MNIST, a modified version of MNIST proposed by Castro et al. (2019), to showcase how deep backtracking contrasts with its interventional counterpart (Pawlowski et al., 2020). The data set consists of three variables, two scalars and an MNIST image. The first scalar variable $T$ describes thickness, whereas the second variable $I$ describes intensity. They have a non-linear relationship and are positively correlated, as can be seen in Fig. 4 **(b)** and **(c)**, where the observational density of thickness and intensity is shown in blue. The known causal relationship between thickness and intensity is depicted in Fig. 4 and we show the true structural equations in App. C. We first train a normalizing flow for thickness and one for intensity (conditionally on thickness) and model the image via a conditional $\beta$-VAE (Higgins et al., 2017). We use $d_i(u'_i, u_i) = \|u'_i - u_i\|_2^2$ as the distance function for DeepBC. We show further experiments using weighted distances in App. D.1.

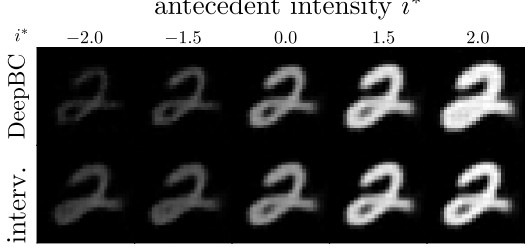

Figure 3: **Counterfactual Images**. DeepBC (top row) changes intensity alongside thickness, since their causal relation is preserved. Interventional counterfactuals (bottom row), on the contrary, solely change the intensity value.

**Results.** Our experiments illustrate distinctive properties of the backtracking approach. For choosing intensity as the antecedent, backtracking preserves causal laws and thus changes thickness in accordance with the change in intensity, creating counterfactuals that resemble the images in the

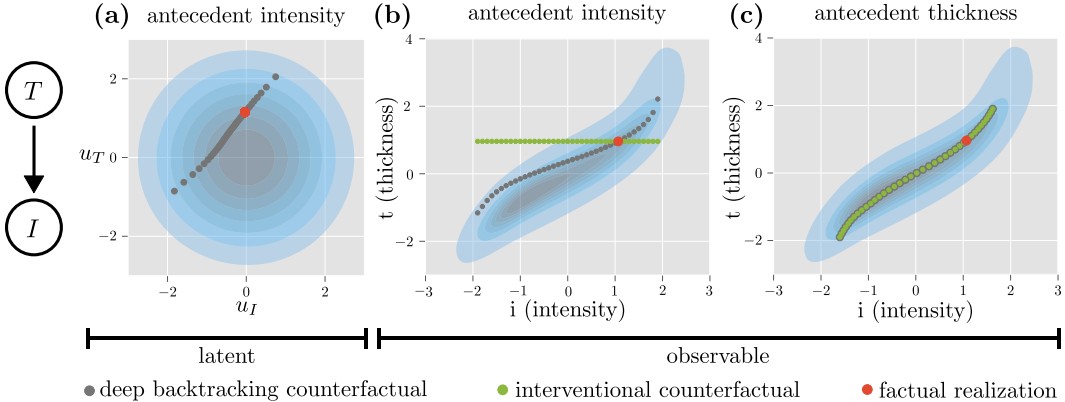

Figure 4: The blue shaded areas indicate probability density on the observed data set and we note that both thickness and intensity variables are causes of the image pixels. **(a)** For varying values of the antecedent $i^*$, both $u_I^*$ and upstream variable $u_T^*$ change (latent variables). Only the deep back-tracking solution is shown. **(b)** Interventional counterfactuals, in contrast to backtracking counter-factuals, leave $t^*$ unchanged for antecendent intensity. **(c)** For antecedent thickness, counterfactual and backtracking counterfactuals are identical.

data set (see Dominguez-Olmedo et al. (2023)), where thickness and intensity change simultane-ously. This is in contrast to the interventional approach, which always leaves thickness unchanged by breaking the causal relationship between both variables. This can be considered a weakness in regard to generating counterfactuals that yield faithful insights into the structural relationships of the data. We show the generated images in Fig. 3. DeepBC arrives at these counterfactuals, since $i^* \neq i$ can either be achieved by choosing a different $u_I^* \neq u_I$ or by changing the upstream $u_T^* \neq u_T$. This is true because $i^*$ also depends on the realization $t^*$, which, in turn, depends on $u_T^*$. As to mini-mize the sum of squares $d_T(u_T, u_T^*) + d_I(u_I, u_I^*)$, DeepBC dissociates both latent variables from their factual realizations, as can be seen in Fig. 4 **(a)**. This entails that the upstream $t^*$ diverges from $t$, which lies in stark contrast to the interventional approach that always keeps upstream variables unmodified (see the bottom row in Fig. 3 and the green dots in Fig. 4 **(b)**).

However, interventional and deep backtracking counterfactuals can also be identical, as visible in Fig. 4 **(c)**, where the thickness variable $T$ is used as antecedent. If the antecedent is a root node of the causal graph $G$, which is the case for $T$, the change in $t^* \neq t$ cannot be traced back to any latent variable other than $u_T$, which is why both $u_I^* = u_I$ and $u_{\text{Img}}^* = u_{\text{Img}}$, analogously to interventional counterfactuals. The change in the value $i^*$ as a function of $t^*$ then solely corresponds to the causal effect of $t^*$, for both counterfactuals (Fig. 4 **(c)**).

## 4.2 CELEBA

**Experimental Setup.** We generate counterfactual celebrity images on the CelebA data set (Liu et al., 2015) with a resolution of $128 \times 128$ using binary attributes with the causal graph as assumed by Yang et al. (2021b). The causal graph is shown in Fig. 5 **(a)**. Our optimization algorithms assume differentiability of $\mathbf{F}$ in $\mathbf{u}$ (§ 3.3), which is why we preprocess the data to use the standardized logits of classifiers that were trained to predict each attribute from the image. Then, analogously to § 4.1, we train a conditional normalizing flow for each attribute and a conditional $\beta$-VAE for the image.

**Baselines & Ablations.** 1) Measuring distance in $\mathbf{x}$: Prior work has measured distance directly in terms of the observable $\mathbf{x}$ rather than latent variables $\mathbf{u}$ that are embedded into a causal model. For the sake of demonstration in Fig. 5, we use a method that encourages sparse solutions in terms of $\mathbf{x}$, akin to Mothilal et al. (2020); Lang et al. (2022). In the style of tabular counterfactual explanations, we train a new regressor, which predicts an attribute from all other attributes (not including the image). We then employ DeepBC or sparse DeepBC on this regressor, but measure distance in the attributes of $\mathbf{x}$ rather than $\mathbf{u}$. 2) Wrong causal graph: We assess how choosing a different causal graph (see Fig. 8 **(e)**) changes the result of the counterfactual. 3) Deep non-causal explanation: According to (6), we use an image regressor ($f_{\hat{Y}}$), together with an *unconditional* auto-encoder ($f_X$) to generate counterfactual explanations. This corresponds to how Jacob et al. (2022); Rodríguez et al. (2021) obtain counterfactual explanations for image data. We show visual comparisons to 2) and 3) in App. D.2 only.

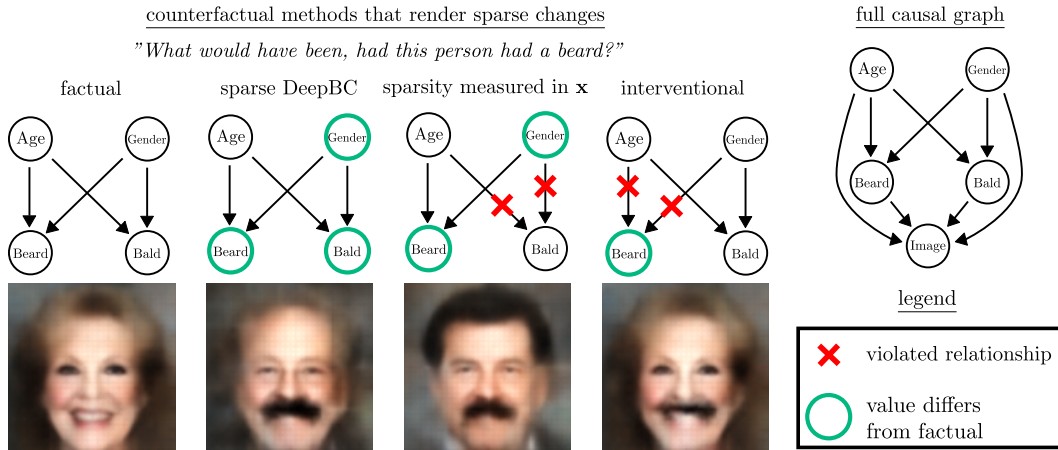

Figure 5: **DeepBC for CelebA**. Both sparse DeepBC and the endogenous sparsity method alter gender to add a beard while keeping age unchanged. Only sparse DeepBC respects the causal downstream: baldness increases as gender changes. In contrast, the method measuring sparsity in **x** leaves the variable bald unchanged, thereby violating the causal relationship.

Table 1: Three considered metrics of the different baselines for 500 iterations. We use squared Euclidean distances for all approaches. Best is in bolt. More details can be found in App. C.1.

|  | distance measured in **x** | interventional | deep non-causal | wrong graph | DeepBC |
|---|---|---|---|---|---|
| plausible | 1.887±2.548 | 1.212±0.996 | **0.933**±0.823 | 1.083±0.904 | 1.045±0.825 |
| obs | 1.503±1.657 | **0.644**±0.953 | 1.427±1.351 | 0.888±1.271 | 0.769±1.066 |
| causal | 1.806±2.280 | 0.757±1.108 | 1.707±1.454 | 0.608±0.967 | **0.559**±0.858 |

**Quantitative Evaluation Metrics.** We evaluate three metrics: plausibility, observational closeness and causal compliance[6]. We define these as

$$\text{plausible}(\mathbf{x}^*) := \sum_{\text{Attr}} \left\| f_{\text{Attr}}^{-1}(\mathbf{x}^*_{\text{pa(Attr)}}, x^*_{\text{Attr}}) \right\|_2^2 / n, \qquad \text{obs}(\mathbf{x}, \mathbf{x}^*) := \sum_{\text{Attr}} \left\| x_{\text{Attr}} - x^*_{\text{Attr}} \right\|_2^2 / n,$$

$$\text{and} \quad \text{causal}(\mathbf{x}, \mathbf{x}^*) := \sum_{\text{Attr}} \left\| f_{\text{Attr}}^{-1}(\mathbf{x}_{\text{pa(Attr)}}, x_{\text{Attr}}) - f_{\text{Attr}}^{-1}(\mathbf{x}^*_{\text{pa(Attr)}}, x^*_{\text{Attr}}) \right\|_2^2 / n,$$

for Attr ∈ {Age, Beard, Gender, Bald} and $n = 4$. We elaborate on the reasoning and detailed implementation of these metrics in App. C.1.

**Results.** Sparse DeepBC measures distance in terms of **u** (subject to the causal laws) rather than **x**. Fig. 5 shows sparse DeepBC ($M = 2$, see § 3.3) and other approaches that are able to generate counterfactuals that render sparse changes with respect to the considered attributes. As can be seen from the causal graph, the elderly woman from the factual image could develop a beard by changing gender and age. Both the endogenous sparsity method and sparse DeepBC choose only gender (it is much more dependent on beard than on age), leaving the value of age fixed. For sparse DeepBC, despite the latent variable $u_{\text{Bald}}$ not being updated, the realization of bald is automatically modified as a downstream effect as encoded by the structural causal model (being old and male often leads to baldness). This lies in contrast to measuring sparsity in terms of **x** directly, where this causal relationship is not taken into account. As a result, the factual value of bald is kept unchanged. We demonstrate further aspects of DeepBC such as modularity and multivariable antecedents in App. D.2.

## 5 RELATED WORK

**Causality in Counterfactual Explanations.** One line of work focuses on the setting of explaining the prediction of a machine learning model based on features along with a graph in the sense of

---

[6]We note that counterfactuals cannot be validated, because ground truth do not exist. The purpose and quality of counterfactuals depend on the application domain and a universal metric does not exist.

(2) (Ying et al., 2019; Bajaj et al., 2021; Lucic et al., 2022; Ma et al., 2022). These prior works alter the graph structure in a fashion that corresponds to neither interventional nor backtracking counterfactuals. Another line of work raises the importance of causality to ensure actionability of counterfactual explanations in the sense that an alternative outcome could have been achieved by performing alternative actions, without violating causal relationships (Karimi et al., 2020; 2021). These works fundamentally differ from ours in that actions break causal relationships, which lies in stark contrast to the backtracking approach.

**Counterfactuals in Deep Structural Causal Models.** The integration of deep generative components such as normalizing flows and variational auto-encoders into structural causal models can be traced back to works from Kocaoglu et al. (2018); Goudet et al. (2018); Pawlowski et al. (2020) and others. Other recent works have explored the use of graph neural networks (Sanchez-Martin et al., 2022), normalizing flows (Khemakhem et al., 2021; Javaloy et al., 2023) or diffusion models (Sanchez & Tsaftaris, 2022) to construct structural causal models. A more detailed review of the literature is included in App. E.

## 6 DISCUSSION

**Identifiablity of Structural Equations.** In general, if the causal graph is not known, neither the structural equations nor the reduced-form (1) that is required for DeepBC can be identified from data (Karimi et al., 2020; Hyvärinen et al., 2023; Locatello et al., 2019). Recent works have established numerous conditions under which identifiability holds (e.g., Lachapelle et al. (2022); Buchholz et al. (2023); Lippe et al. (2023)). Beyond that, recent work has shown that interventional counterfactuals can be identified for invertible structural causal models (Nasr-Esfahany et al., 2023) and it could be that similar results hold for backtracking as well.

**Sampling Counterfactuals from a Distribution.** Prior work has raised the importance of obtaining multiple and diverse explanations for a single example (Mothilal et al., 2020), which our method currently only allows by varying the choice of distance functions $d_i$ in (3). As to fulfill this objective, it has been suggested to sample counterfactuals from a probability distribution (Guidotti, 2022, §3.1). Prior methods have explored the use of amortized inference to obtain distributions over counterfactual explanations (Mahajan et al., 2019). However, this approach did not yield satisfactory results in the context of our method, because the true underlying latent posterior (see App. A.1) is complex. Yet, a possible way forward may be to consider approaches such as flow-based models (Kingma et al., 2016) or semi-amortization (Kim et al., 2018).

**Non-Invertible Generative Models.** A possible future line of research could be to explore how backtracking could be implemented for generative models whose latent variables cannot be inferred deterministically from the factual realization, such as diffusion models (Ho et al., 2020) and generative adversarial networks (Goodfellow et al., 2014), both of which are not invertible in general. One conceivable solution might be to adapt (4) as to jointly optimize over $\mathbf{u}$ and $\mathbf{u}^*$ in the latent space.

**Non-Causal Counterfactual Explanations.** The explicit access to a causal model allows for its versatility, modularity and the capability to obtain causally compliant solutions for varying choices of distance functions (§ 3.2, § 4). We note however that non-causal methods that do not rely on knowledge of the (non-identifiable) reduced-form can yield results of similar appearance in some settings (see Fig. 9 **(d)** in App. D.2).

## 7 CONCLUSION

In this work, we presented DeepBC, a practical algorithm for computing backtracking counterfactuals for deep structural causal models. We compared DeepBC to interventional counterfactuals and the main formulations employed in the field of counterfactual explanations. We found that compared to prior work in counterfactual explanations, DeepBC is versatile in that it supports complex graph structures, compliant with the given causal model and modular in that it enables generalization to out-of-domain settings. In fact, DeepBC can be seen as a general method for computing counterfactual explanations that measures distances between factual and counterfactual in the structured latent space of a causal model. We empirically demonstrated the merits of our approach in comparison to prior work, where we highlight the importance of taking causal relationships into account.

REPRODUCIBILITY STATEMENT

Our anonymized source code is available at https://anonymous.4open.science/r/DeepBC_REVISED-2160. The instructions for reproducing all visualizations are provided in the README.md file at the top level of the repository. All parameters can be found in the config folders within the respective subfolders. In addition, we provide a detailed description of the optimization parameters in App. B.1, training procedures in App. B.2.1 and deep learning architectures in App. B.2.2.

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

# A FORMALISMS & DERIVATIONS

## A.1 FORMAL DEFINITION OF INTERVENTIONAL AND BACKTRACKING COUNTERFACTUALS

Both kinds of counterfactuals can be computed in a three-step-procedure.

**Interventional Counterfactuals**

1. **Abduction**: Compute the distribution of $\mathbf{U} \mid \mathbf{x}$, given the factual realization $\mathbf{x}$ of $\mathbf{X}$.

2. **Action**: Obtain an altered collection of structural assignments $(f_1^*, f_2^*, ..., f_n^*)$ by setting $x_i \leftarrow x_i^* = f_i^*$, for all $i \in S$. Leave all other structural assignments unmodified, i.e., $f_j^* = f_j$, for all $j \notin S$.

3. **Prediction**: Compute a distribution over $\mathbf{X}_I^*$ as the pushforward of the distribution of $\mathbf{U} \mid \mathbf{x}$ by $\mathbf{F}^*$.

**Backtracking Counterfactuals**

1. **Cross-World Abduction**: Use the antecedent $\mathbf{x}_S^*$ and the factual realization $\mathbf{x}$ to obtain $p(\mathbf{u}^*, \mathbf{u} \mid \mathbf{x}_S^*, \mathbf{x})$, using the backtracking conditional $p(\mathbf{u}^*|\mathbf{u})$ and latent prior density $p(\mathbf{u})$:

$$p(\mathbf{u}^*, \mathbf{u} \mid \mathbf{x}_S^*, \mathbf{x}) \; = \; \frac{p(\mathbf{u}^*, \mathbf{u}, \mathbf{x}_S^*, \mathbf{x})}{p(\mathbf{x}_S^*, \mathbf{x})} \; = \; \frac{p(\mathbf{u}^*|\mathbf{u})\, p(\mathbf{u})\, \delta_\mathbf{x}(\mathbf{F}(\mathbf{u}))\delta_{\mathbf{x}_S^*}(\mathbf{F}_S(\mathbf{u}^*))}{\int \int p(\mathbf{u}^*|\mathbf{u})\, p(\mathbf{u})\, \delta_\mathbf{x}(\mathbf{F}(\mathbf{u}))\delta_{\mathbf{x}_S^*}(\mathbf{F}_S(\mathbf{u}^*))\, d\mathbf{u}\, d\mathbf{u}^*},$$

where $\delta_\mathbf{x}(\,\cdot\,)$ refers to the dirac delta at $\mathbf{x}$.

2. **Marginalization**: Marginalize over $\mathbf{U}$ to obtain the density $p(\mathbf{u}^* \mid \mathbf{x}_S^*, \mathbf{x})$ of the counterfactual posterior:

$$p(\mathbf{u}^* \mid \mathbf{x}_S^*, \mathbf{x}) \; = \; \int p(\mathbf{u}^*, \mathbf{u} \mid \mathbf{x}_S^*, \mathbf{x})\, d\mathbf{u}.$$

3. **Prediction**: Compute a distribution over $\mathbf{X}_B^*$ by marginalizing over the counterfactual latents $\mathbf{U}^*$:

$$p(\mathbf{x}^* \mid \mathbf{x}_S^*, \mathbf{x}) \; = \; \int p(\mathbf{u}^* \mid \mathbf{x}_S^*, \mathbf{x})\delta_{\mathbf{x}^*}(\mathbf{F}(\mathbf{u}^*))\, d\mathbf{u}^*.$$

## A.2 FORMAL DERIVATION OF DEEPBC

We derive (3) from the three-step-procedure of backtracking counterfactuals (see App. A.1) as follows:

1. **Cross-World Abduction**: By the deterministic relationship between latents and observables, we see that

$$\begin{aligned} p(\mathbf{u}^*, \mathbf{u} \mid \mathbf{x}_S^*, \mathbf{x}) \; &= \; p(\mathbf{u}^* \mid \mathbf{u}, \mathbf{x}_S^*, \mathbf{x})\, p(\mathbf{u} \mid \mathbf{x}_S^*, \mathbf{x}) \; = \; p(\mathbf{u}^* \mid \mathbf{u}, \mathbf{x}_S^*)\, p(\mathbf{u} \mid \mathbf{x}) \\ &= \; p(\mathbf{u}^* \mid \mathbf{u}, \mathbf{x}_S^*)\, \delta_{\mathbf{F}^{-1}(\mathbf{x})}(\mathbf{u}). \end{aligned}$$

2. **Marginalization**: All the probability is located at $\mathbf{F}^{-1}(\mathbf{x})$, which is why marginalization reduces to

$$p(\mathbf{u}^* \mid \mathbf{x}_S^*, \mathbf{x}) \; = \; p(\mathbf{u}^*, \mathbf{u} = \mathbf{F}^{-1}(\mathbf{x}) \mid \mathbf{x}_S^*, \mathbf{x}).$$

3. **Prediction**: By the deterministic relationship between latents and observables, we obtain samples from $\mathbf{X}^* \mid \mathbf{x}_S^*, \mathbf{x}$ simply by sampling from $\mathbf{U}^* \mid \mathbf{F}^{-1}(\mathbf{x}), \mathbf{x}_S^*$ and then subsequently mapping these samples through the function $\mathbf{F}(\mathbf{u}^*)$ to obtain the corresponding observables $\mathbf{x}^*$:

$$\mathbf{u}^* \; \sim \; \mathbf{U}^* \mid \mathbf{F}^{-1}(\mathbf{x}), \mathbf{x}_S^*, \quad \mathbf{x}^* \; = \; \mathbf{F}(\mathbf{u}^*).$$

Instead of sampling, however, we restrict ourselves to the mode of the distribution of $\mathbf{U}^* \mid \mathbf{F}^{-1}(\mathbf{x}), \mathbf{x}_S^*$. We assume that the backtracking conditional density $p(\mathbf{u}^*|\mathbf{u})$ has the following form

$$p(\mathbf{u}^*|\mathbf{u}) \; \propto \; \exp\left\{ -\sum_{i=1}^n d_i(u_i^*, u_i) \right\},$$

where $d$ is a distance function. Then, we have

$$p(\mathbf{u}^* \mid \mathbf{F}^{-1}(\mathbf{x}), \mathbf{x}_S^*) \; \propto \; \begin{cases} \exp\left\{ -\sum_{i=1}^n d_i\left(u_i^*, \mathbf{F}_i^{-1}(\mathbf{x})\right) \right\}, & \text{if } \mathbf{F}_S(\mathbf{u}^*) \; = \; \mathbf{x}_S^* \\ 0, & \text{otherwise.} \end{cases}$$

By taking the logarithm and ignoring constants, we obtain

$$\log p(\mathbf{u}^* \mid \mathbf{F}^{-1}(\mathbf{x}), \mathbf{x}_S^*) \; = \; \begin{cases} -\sum_{i=1}^n d_i\left(u_i^*, \mathbf{F}_i^{-1}(\mathbf{x})\right), & \text{if } \mathbf{F}_S(\mathbf{u}^*) \; = \; \mathbf{x}_S^* \\ -\infty, & \text{otherwise.} \end{cases}$$

We conclude by noting that $\arg\max_{\mathbf{u}^*} \log p(\mathbf{u}^* \mid \mathbf{F}^{-1}(\mathbf{x}), \mathbf{x}_S^*)$, composed with $\mathbf{F}$, is equivalent to (3).

### A.3 DERIVATION OF (3)

As a result of the linearization of $\mathbf{F}$, (7) simplifies to

$$(\mathbf{u}' - \mathbf{u})^\top \mathbf{W}(\mathbf{u}' - \mathbf{u}) \; + \; \lambda ||\mathbf{J}_S(\mathbf{u}' - \mathbf{u}) + \mathbf{F}_S(\mathbf{u}) - \mathbf{x}_S^*||_2^2$$
$$= \; (\mathbf{u}' - \mathbf{u})^\top \mathbf{W}(\mathbf{u}' - \mathbf{u}) \; + \; \lambda ||\mathbf{J}_S\mathbf{u}' - \tilde{\mathbf{x}}_S^*||_2^2 \; =: \; \tilde{\mathcal{L}}(\mathbf{u}'). \quad (9)$$

We see that $\tilde{\mathcal{L}}(\mathbf{u}')$ is convex and differentiable with respect to $\mathbf{u}'$, which means that $\nabla_{\mathbf{u}'}\tilde{\mathcal{L}}(\mathbf{u}') = \mathbf{0}$ implies optimality of $\mathbf{u}'$. To derive $\mathbf{u}'_{\text{opt}}$, we observe that

$$\nabla_{\mathbf{u}'}\tilde{\mathcal{L}}(\mathbf{u}') \; = \; 2(\mathbf{W}(\mathbf{u}' - \mathbf{u}) + \lambda \mathbf{J}_S^\top \mathbf{J}_S \mathbf{u}' - \mathbf{J}_S^\top \tilde{\mathbf{x}}_S^*).$$

As a result, $\mathbf{u}'_{\text{opt}}$ is given by

$$\mathbf{u}'_{\text{opt}} \; = \; (\mathbf{W} + \lambda \mathbf{J}_S^\top \mathbf{J}_S)^{-1}(\mathbf{W}\mathbf{u} + \lambda \mathbf{J}_S^\top \tilde{\mathbf{x}}_S^*).$$

## B IMPLEMENTATION

### B.1 TECHNICAL DETAILS AND COMMENTS FOR THE DEEPBC OPTIMIZATION ALGORITHM

In practice, we implement (8) as follows

$$\hat{\mathbf{u}}^* = (\lambda^{-1}\mathbf{W} + \mathbf{J}_S^\top \mathbf{J}_S)^\dagger (\lambda^{-1}\mathbf{W}\mathbf{u} + \mathbf{J}_S^\top \tilde{\mathbf{x}}_S^*), \quad (10)$$

where $\dagger$ denotes Moore-Penrose pseudoinverse. We employ (10) rather than (8) for the reason of numerical stability. The main computational bottleneck in Alg. 1 is the computation of the pseudoinverse $(\lambda^{-1}\mathbf{W} + \mathbf{J}_S^\top \mathbf{J}_S)^\dagger$ in (10), which comes at a cost of $\mathcal{O}(\#\text{it} \cdot \dim(\mathbf{u})^3)$, compared to $\mathcal{O}(\#\text{it} \cdot \dim(\mathbf{u}))$ for gradient descent. We note, however, that the dimensionality of the latent space is typically not very large in our experiments. The maximum dimension is 516 for CelebA, due to 4 attributes and 512-dimensional latent space of the VAE. We also stress that $\mathbf{J}_S$ is sparse (many 0 entries) when $S$ covers attribute variables, because the 512-dimensional latent vector is not upstream of any attribute. We do not run experiments where many variables are upstream of the antecedent variable and stress that this may affect the performance of Alg. 1.

In our experiments, we find Alg. 1 to converge much more quickly, as can be seen in Fig. 6. Typically, convergence can be expected to occur within $\approx 5$ iterations, while fulfilling the constraint realiably (see table in Fig. 6). When applying gradient-based methods like Adam instead of our approach, we observe that the convergence rate is sensitive to the choice of learning rate. The plot for $\lambda = 10^6$ shows that the linearization method can lead to oscillations if $\lambda$ is chosen too large, which likely stems from small eigenvalues of $\lambda^{-1}\mathbf{W} + \mathbf{J}_S^\top \mathbf{J}_S$ (we not that $\mathbf{J}_S^\top \mathbf{J}_S$ is low-rank) that give rise to numerical issues. However, these oscillations can be detected early on. Similar to Levenberg-Marquardt, we could include a small damping variable $\epsilon > 0$ to alleviate this issue. We would then arrive at

$$\hat{\mathbf{u}}^* = (\lambda^{-1}\mathbf{W} + \mathbf{J}_S^\top \mathbf{J}_S + \mathbf{I}\epsilon)^\dagger (\lambda^{-1}\mathbf{W}\mathbf{u} + \mathbf{J}_S^\top \tilde{\mathbf{x}}_S^*).$$

We do not explore this possibility in the present work as we do not encounter these issues in our experiments.

Adam's convergence highly depends on the choice of learning rate. We suspect that this is due to the poorly conditioned Hessian that comes from choosing large $\lambda$. However, large $\lambda$ is required in

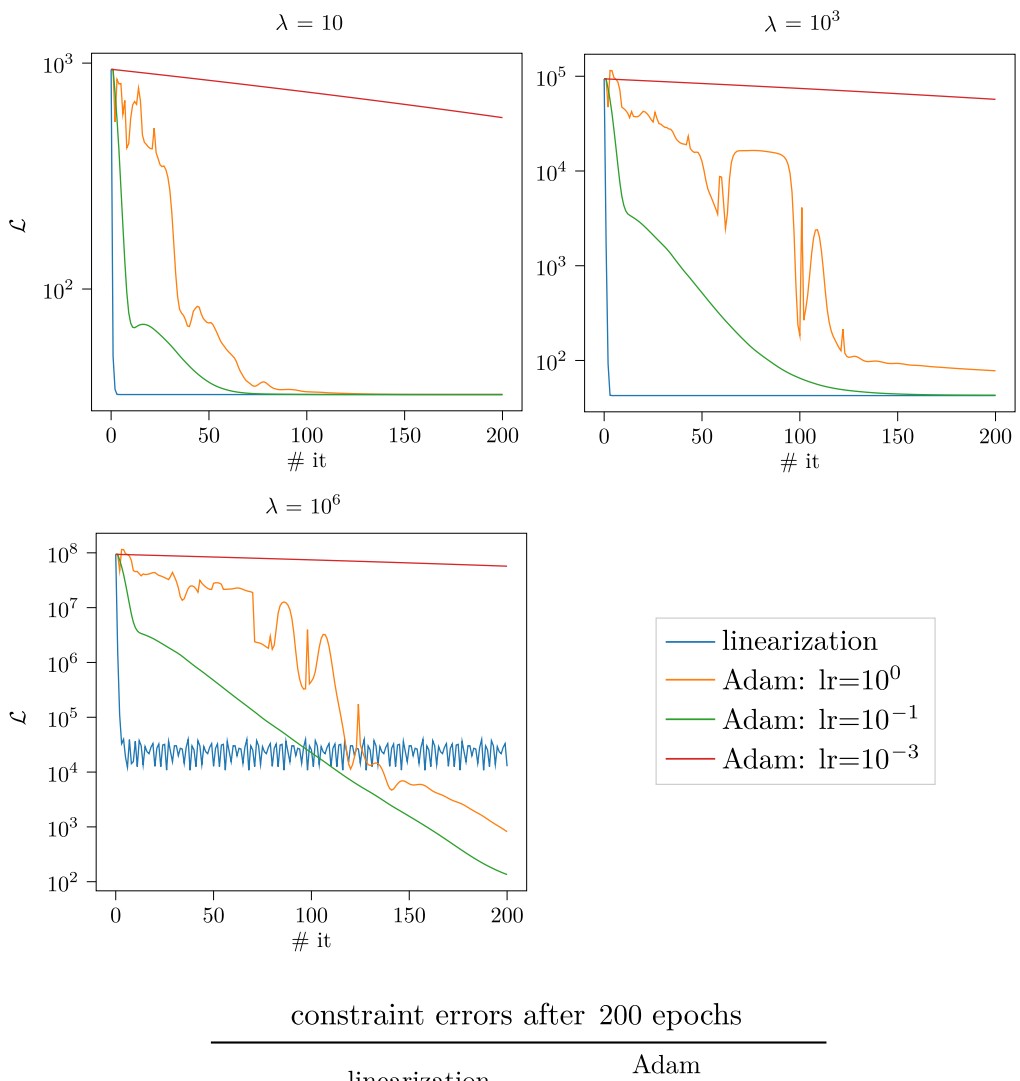

constraint errors after 200 epochs

|  | linearization | Adam | | |
| --- | --- | --- | --- | --- |
|  |  | $lr = 10^0$ | $10^{-1}$ | $10^{-3}$ |
| $\lambda = 10$ | 0.412 | 0.412 | 0.412 | 56.978 |
| $\lambda = 10^3$ | 0.000 | 0.000 | 0.000 | 56.895 |
| $\lambda = 10^6$ | 0.012 | 0.000 | 0.000 | 56.894 |

Figure 6: The figures show the sum of penalty losses (7) over all points in Fig. 4 **(b)** for optimizing it for 200 iterations on a $\log_{10}$ scale. Comparison of the Adam optimizer with various learning rates in comparison to constraint linearization (Alg. 1) for different choices of penalty parameter $\lambda$. The table shows the sum of constraint errors $\|\mathbf{F}_S(\mathbf{u}') - \mathbf{x}_S^*\|_2^2$ after 200 iterations (How well the constraints are fulfilled).

order to (at least approximately) fulfill the constraint in (3) (see top row in the table of Fig. 6).

In our experiments, we always use DeepBC via constraint linearization (Alg. 1) with $\lambda = 10^3$ and $\#\text{it} = 30$. $\lambda$ is chosen empirically and our choice yielded convincing results in all experiments. The choice of iteration number is a conservative upper bound for the algorithm.

## B.2 IMPLEMENTATION DETAILS OF THE DEEP STRUCTURAL CAUSAL MODELS

For all experiments, we use `PyTorch` (Paszke et al., 2019), `PyTorch Lightning` (Falcon, William and The PyTorch Lightning team, 2019) and `normflows` (Stimper et al., 2023).

### B.2.1 TRAINING PROCEDURES

We train all models with the following parameters:

| optimizer | train/val. split ratio | regularization | max. # epochs |
|-----------|------------------------|----------------|---------------|
| Adam | 0.8 | early stopping | 1000 |

**Morpho-MNIST.** We use the same training parameters for both normalizing flow models. Patience refers to the number of epochs without further decrease in validation loss that early stopping regularization waits.

| model | batch size train | batch size val. | learning rate | patience |
|-------|------------------|-----------------|---------------|----------|
| **Flow** | 64 | full | $10^{-3}$ | 2 |
| **VAE** | 128 | 256 | $10^{-6}$ | 10 |

**CelebA.** We use the same training parameters for all normalizing flow models.

| model | batch size train | batch size val. | learning rate | patience |
|-------|------------------|-----------------|---------------|----------|
| **Flow** | 64 | 256 | $10^{-3}$ | 2 |
| **VAE** | 128 | 256 | $10^{-6}$ | 50 |

### B.2.2 NETWORK ARCHITECTURES

**Notation.** We denote concatenations of variables by $[\cdot, \cdot, ..., \cdot]$. We denote modules that are repeated $n$ times by a superscript $(n)$. For instance, $\text{Linear}^{(2)}(u)$ is shorthand for $\text{Linear} \circ \text{Linear}(u)$, i.e., two linear layers.

**Flow Layers.** In all of our experiments, we make use of common types of flow layers:

$\text{QuadraticSpline}(u_i)$ is a standard quadratic spline flow (Durkan et al., 2019).

$\text{ConstScaleShift}(u_i)$ performs a constant affine transformation with learned, but unconditional, location and scale parameters $\mu$ and $\sigma$:

$$\text{ConstScaleShift}(u_i) = \sigma \cdot u_i + \mu.$$

$\text{ScaleShift}(u_i, \mathbf{x}_{\text{pa}(i)})$ performs the same operation as $\text{ConstScaleShift}(u_i)$, but $\mu$ and $\sigma$ are computed as a function of $u_i$ and $\mathbf{x}_{\text{pa}(i)}$ via a two-layer Masked Autoencoder for Distribution Estimation (MADE) module (Germain et al., 2015) with ReLU activation functions and one-dimensional hidden units.

**Morpho-MNIST.** For the thickness variable, we construct the flow as

$$f_T(u_T) = \text{ConstScaleShift} \circ \text{QuadraticSpline}^{(5)}(u_T).$$

For intensity, we use

$$f_I(t, u_I) = \text{ConstScaleShift} \circ \text{Sigmoid} \circ \text{QuadraticSpline}^{(3)} \circ \text{ScaleShift}([t, u_I]),$$

where Sigmoid denotes the (constant) sigmoid function.

For the MNIST image, we use a convolutional $\beta$-VAE (Higgins et al., 2017) with $\beta = 3$ and the following encoder parameterization:

$$f_{\text{Img}}(t, i, \text{img}) \approx e_{\text{Img}}(t, i, \text{img})$$
$$= \text{Linear}\left(\left[t,\ i,\ \left(\text{Linear} \circ \text{Pool2D} \circ (\text{ReLU} \circ \text{Conv2D})^{(4)}\right)(\text{img})\right]\right),$$

where the Conv2D layers (starting with parameters from the layer closest to the input) are parameterized by `out_channels` $= (8, 16, 32, 64)$, `kernel_size` $= (4, 4, 4, 3)$, `stride` $= (2, 2, 2, 2)$, `padding` $= (1, 1, 1, 0)$. The linear layers are analogously parameterized with the output dimensions `out` $= (128, 16, 16)$, i.e., $\dim(u_{\text{Img}}) = 32$. For the decoder, we use

$$f_{\text{Img}}^{-1}(t, i, u_{\text{Img}}) \approx d_{\text{Img}}(t, i, u_{\text{Img}})$$
$$= \text{TransConv2D} \circ (\text{ReLU} \circ \text{TransConv2D})^{(4)} \circ \text{Linear}\left([t, i, u_{\text{Img}}]\right),$$

where the linear layer has output dimension `out` $= 64$ and the transpose convolution layers (starting with parameters from the layer closest to the input) are parameterized by `out_channels` $= (64, 32, 16, 1)$, `kernel_size` $= (3, 4, 4, 4)$, `stride` $= (2, 2, 2, 2)$, `padding` $= (0, 1, 0, 1)$.

**CelebA.** We preprocess all attributes via separate classifiers $C_{\text{Attr}}$, i.e., one individual classifier per attribute. The classifier has the following architecture:

$$C_{\text{Attr}}(\text{img}) = \text{Linear} \circ \text{Dropout} \circ \text{ReLU} \circ \text{Linear} \circ (\text{MaxPool2D} \circ \text{ReLU} \circ \text{Conv2D})^{(4)}(\text{img}). \quad (11)$$

We then standardize the output logits of $C_{\text{Attr}}$, for each attribute individually.

As for MorphoMNIST, we train one normalizing flow for each attribute. For this, we use the standardized logits from the classifiers rather than the original binary attributes from the data set. To model the non-Gaussian distributions, we employ the following flow architecture:

$$f_{\text{Attr}}(t, u_{\text{Attr}}) = \text{ScaleShift}\left(\left[\left(\text{QuadraticSpline}^{(10)} \circ \text{ConstScaleShift}\right)(u_{\text{Attr}}),\ \mathbf{x}_{\text{pa(Attr)}}\right]\right),$$

For the $\beta$-VAE with $\beta = 3$, we follow a slightly different approach as for B.2.2. Rather than concatenating the conditional variables $\mathbf{x}_{\text{pa}(i)}$ at the end of the encoder, we instead create an additional channel $\text{chan}_{\text{attr}}$ for each attribute attr that we concatenate to the RGB channels of the image. Specifically, we obtain the channel by broadcasting the continuous attribute value $x_{\text{Attr}}$ like

$$\text{ch}_{\text{Attr}} = \mathbf{1}_{128 \times 128} \cdot x_{\text{Attr}},$$

where we replace the MADE module by a linear function for Bald, since the signal-to-noise ratio is low for this variable. The reason is that Beard is the only variable that cannot be modeled well as a linear function of its causal parents Age and Gender.

where $\mathbf{1}_{128 \times 128}$ is a matrix of dimensionality $128 \times 128$ that consists only of 1. We then feed $\tilde{\mathbf{x}} := [x_R, x_G, x_B, \text{ch}_{\text{Beard}}, \text{ch}_{\text{Bald}}, \text{ch}_{\text{Gender}}, \text{ch}_{\text{Age}}] \in \mathbb{R}^{128 \times 128 \times 7}$ directly into the encoder with the following architecture (roughly inspired by Ghosh et al. (2020)):

$$f_{\text{Img}}(\tilde{\mathbf{x}}) \approx e_{\text{Img}}(\tilde{\mathbf{x}})$$
$$= \text{Linear} \circ \text{Pool2D} \circ (\text{ReLU} \circ \text{BatchNorm2D} \circ \text{Conv2D})^{(6)}(\tilde{\mathbf{x}}),$$

where the final linear layer has output dimension `out` $= 512$ and the transpose convolution layers (starting with parameters from the layer closest to the input) are parameterized by `out_channels` $= (128, 128, 128, 256, 512, 1024)$, `kernel_size` $= (3, 3, 3, 3, 3, 3)$, `stride` $= (2, 2, 2, 2, 2, 2)$, `padding` $= (1, 1, 1, 1, 1, 1)$. For the decoder, noting that $\mathbf{x}_{\text{pa(Img)}} = [x_{\text{Beard}}, x_{\text{Bald}}, x_{\text{Gender}}, x_{\text{Age}}]$, we use

$$f_{\text{Img}}^{-1}(\mathbf{x}_{\text{pa(Img)}}, u_{\text{Img}}) \approx d_{\text{Img}}(\mathbf{x}_{\text{pa(Img)}}, u_{\text{Img}})$$
$$= \text{TransConv2D} \circ (\text{ReLU} \circ \text{BatchNorm2D} \circ \text{TransConv2D})^{(4)} \circ \text{Linear}\left([\mathbf{x}_{\text{pa(Img)}}, u_{\text{Img}}]\right),$$

where the first linear layer maps to $\mathbb{R}^{4 \cdot 1024}$, which is then reshaped to a feature map in $\mathbb{R}^{2 \times 2 \times 1024}$. The consecutive transposed convolutional layers have the parameters `out_channels` $= (512, 256, 128, 128, 128)$, `kernel_size` $= (3, 3, 3, 3, 3)$, `stride` $= (2, 2, 2, 2, 2)$, `padding` $= (1, 1, 1, 1, 1)$.

## C  GROUND TRUTH STRUCTURAL EQUATIONS MORPHO-MNIST

The structural equation for thickness $T$ and intensity $I$ are given as

$$T \leftarrow 0.5 + U_T, \qquad\qquad\qquad\qquad U_T \sim \Gamma(10, 5)$$
$$I \leftarrow 191 \cdot \text{Sigmoid}\,(0.5 \cdot U_I + 2 \cdot T - 5) \,+\, 64, \qquad U_I \sim \mathcal{N}(0, 1).$$

For details about how the MNIST images were modified as to change perceived thickness and intensity, we refer the reader to Pawlowski et al. (2020).

### C.1  METRIC IMPLEMENTATIONS

We only evaluate the metrics on the attribute values, because the tabular method (method 1)) does not generate images. Furthermore, this leads to the following choice of distance metric

$$\text{obs}(\mathbf{x}, \mathbf{x}^*) := \sum_{\text{Attr}} \| x_{\text{Attr}} - x_{\text{Attr}}^* \|_2^2 / n, \tag{12}$$

which is adopted from early work on counterfactual explanations (see e.g. Wachter et al. (2017)). We note that $x_{\text{Attr}}$ are standardized logits, so we do not scale the distances. The other metric

$$\text{plausible}(\mathbf{x}^*) := \sum_{\text{Attr}} \left\| f_{\text{Attr}}^{-1}(\mathbf{x}_{\text{pa(Attr)}}^*, x_{\text{Attr}}^*) \right\|_2^2 / n \tag{13}$$

penalizes the deviation of the latent variable $u^*$ from its mean ($u^* = 0$). We can think of this loss intuitively as penalizing the amount of noise that would be necessary to generate the counterfactual values under the assumption that the causal mechanisms are in place. Finally,

$$\text{causal}(\mathbf{x}, \mathbf{x}^*) := \sum_{\text{Attr}} \left\| f_{\text{Attr}}^{-1}(\mathbf{x}_{\text{pa(Attr)}}, x_{\text{Attr}}) - f_{\text{Attr}}^{-1}(\mathbf{x}_{\text{pa(Attr)}}^*, x_{\text{Attr}}^*) \right\|_2^2 / n \tag{14}$$

measures the distance of all latents for the structural model. We note that this loss corresponds to the distance term (the left summand) in (3), restricted to attribute variables and for squared Euclidean norm. Again, we restrict the loss to attributes for fairer comparison to the tabular method. Since we do not have access to ground truth structural equations $(f_1, f_2, ..., f_n)$ in CelebA, we use the ones that were trained on the data set. We furthermore note that incorporating the antecedent variable into the loss is not an issue either, because it is fixed for all methods. For the deep non-causal explanation method, we only obtain the counterfactual image, without explicit access to the attribute variables. In order to extract those, we use the (standardized) logits of classifiers that were trained to predict the attributes from the image.

We obtain the numbers in table § 4.2 as follows:

We sample a factual data point $\mathbf{x} = \mathbf{F}(\mathbf{u}), \mathbf{u} \sim \mathcal{N}(\mathbf{0}, \mathbf{I})$. Then, we sample an attribute uniformly, i.e.

$$\text{attr} \sim \mathcal{U}(\{\text{age, gender, beard, bald}\})$$

and construct the corresponding antecedent as

$$x_{\text{Attr}}^* \sim \mathcal{N}(0, 1).$$

We then compute the counterfactual $\mathbf{x}^*$ to evaluate all three loss function (12), (13) and (14). This process is repeated 500 times. The final reported scores are the arithmetic means over the individual metrics, including $\pm 1$std.

# D   ADDITIONAL EXPERIMENTS

## D.1   MORPHO-MNIST

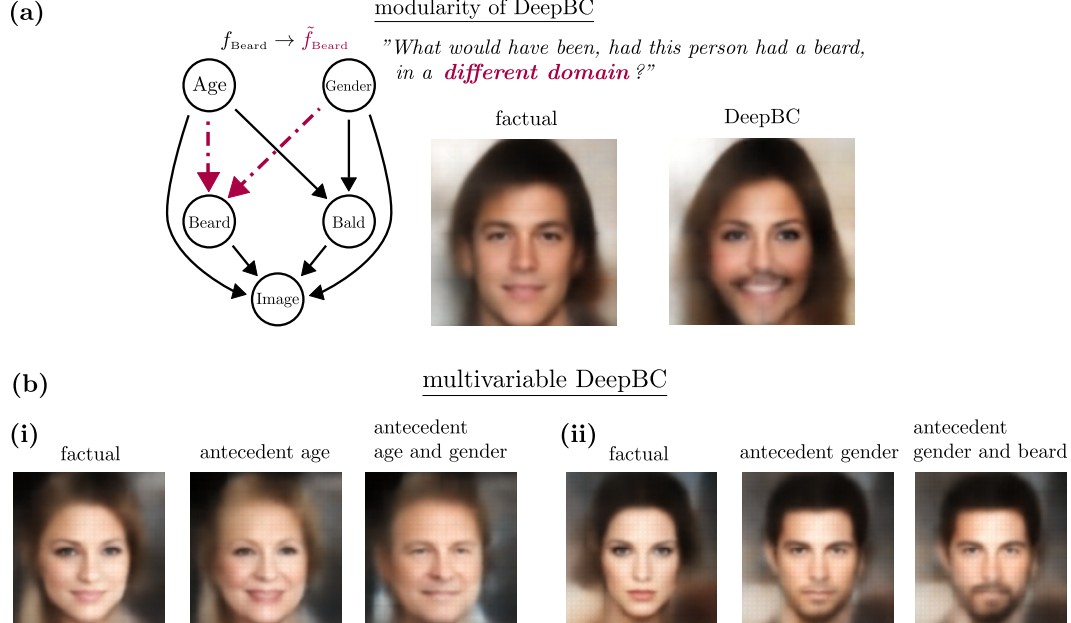

Figure 7: We run Fig. 4 **(b)** multiple times, fixing $w_I = w_{\text{Img}} = 1$ and changing only $w_T$. We see that the backtracking solution approach the interventional solution (see Fig. 4 **(b)**) as we increase $w_T$, thus preserving the value of thickness more as we increase the weight.

## D.2   CELEBA

**(a)**

modularity of DeepBC

$f_{\text{Beard}} \to \tilde{f}_{\text{Beard}}$

"*What would have been, had this person had a beard, in a **different domain**?*"

factual          DeepBC

**(b)**

multivariable DeepBC

**(i)** factual   antecedent age   antecedent age and gender   **(ii)** factual   antecedent gender   antecedent gender and beard

Figure 8: **Additional plots for CelebA, part I).** **(a)** A male, beardless person develops female traits as an upstream of antecedent beard, where the learned structural equation $f_{\text{Beard}}$ is replaced by $\tilde{f}_{\text{Beard}}$ to mimic an out-of-domain setting that can be handled by DeepBC. $\tilde{f}_{\text{Beard}}$ here is constructed manually such that being female is strongly positively correlated with having a beard, unlike in the model that was learned from data. **(b)** Two examples for multivariable DeepBC.

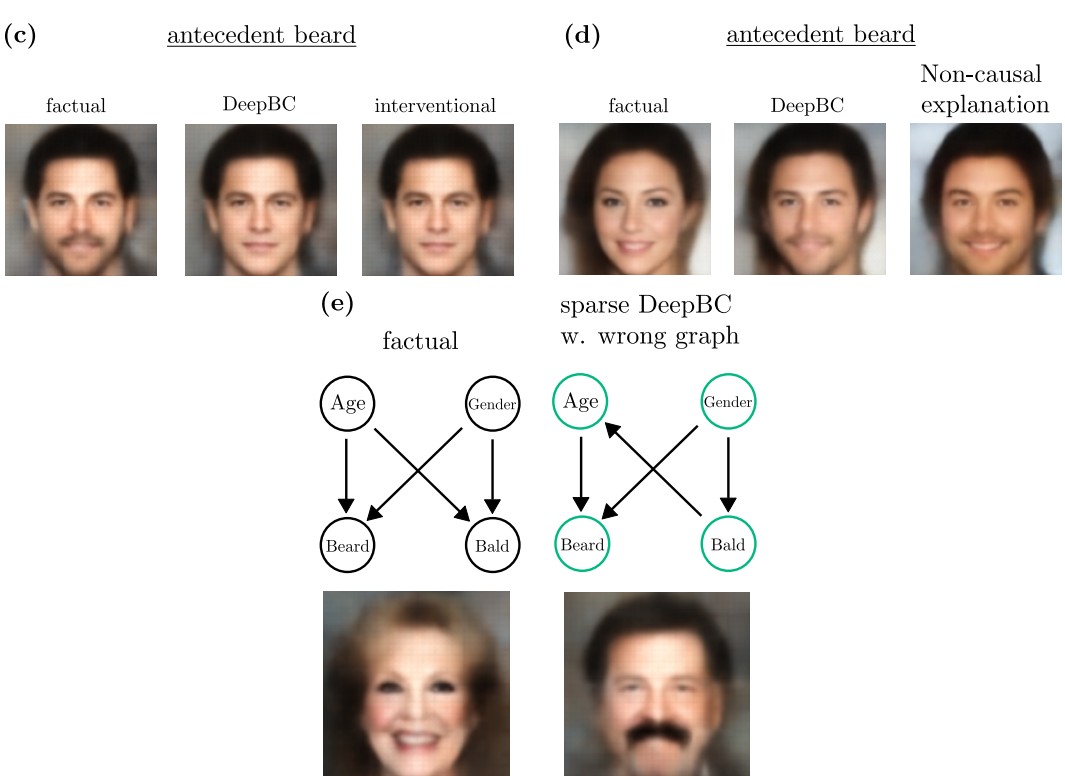

Figure 9: **Additional plots for CelebA, part II).** **(c)** DeepBC takes into account non-deterministic relationships between variables: In this setting, the removed beard is traced back to $u_{\text{Beard}}$ rather than other variables. The result is highly similar to the interventional example (plotted for comparison). **(d)** Non-causal explanation methods based on (6) yield similar results to vanilla DeepBC in some settings. **(e)** The right causal graph shows the wrong graph used to generate this image. It demonstrates that the solution of (sparse) DeepBC is dependent on using the graph structure. Further, it shows that sparse DeepBC does not always generate sparse solutions. This is because downstream variables are always updated.

## E    RELATED WORK (COMPREHENSIVE)

This section is organized into two lines of prior work. The first line encompasses methods that incorporate causality into the field of counterfactual explanations. We do however note that the general field of counterfactual explanations has made many significant advances that are not directly related to causality in recent years. For a comprehensive overview over these developments, we refer to Guidotti (2022) and Verma et al. (2020). The second line discusses how deep neural networks have been used within the context of structural causal models, as to facilitate counterfactual computation.

**Causality in Counterfactual Explanations.** As explained in § 3.1, our DeepBC approach is related to the field of counterfactual explanations. Our work builds therefore on earlier approaches that generate counterfactual explanations by incorporating causal models: One line of work focuses on the setting of explaining the prediction of a machine learning model based on features along with a (causal) graph in the sense of (2) (Ying et al., 2019; Bajaj et al., 2021; Lucic et al., 2022; Ma et al., 2022). Whereas our approach never manipulates the given graph structure of the causal graph, these prior works alter the graph structure in a fashion that corresponds to neither interventional nor backtracking counterfactuals.

Another line of work raises the importance of causality to ensure actionability of counterfactual explanations in a sense that an alternative outcome could have been achieved by performing alternative actions, without violating causal relationships (Karimi et al., 2020; 2021). These works fundamentally differ from ours in that actions break causal relationships, which lies in stark contrast to the backtracking approach. The latter seeks to trace back counterfactuals to changes in latent variables rather than changes in causal relationships. However, this line of research is related to ours in that it argues for respecting causal mechanisms in generating counterfactuals.

The most similar existing work to ours is that of Mahajan et al. (2019). Similarly to the present work, the authors employ deep generative modelling and measure distance between factual and counterfactual examples in a latent space. The most distinctive difference to our work is that Mahajan et al. (2019) impose causal constraints via a *causal proximity loss* in the observable variables $\mathbf{x}$ that assumes additive Gaussian noise. This is in contrast to the backtracking philosophy (Jackson, 1977) that we follow. In our approach, all changes are traced back solely to latent variables $\mathbf{u}$ that are embedded into the deep causal model such that causal constraints are fulfilled automatically. This obviates the need for an additional loss and allows easily for non-additive noise dependencies (see App. C). At the same time, our approach is more versatile as any of the given variables could be used as antecedent, whereas Mahajan et al. (2019) only support a specific label variable.

**Counterfactuals in Deep Structural Causal Models.** The integration of deep generative components such as normalizing flows and variational auto-encoders into structural causal models can be traced back to works from Kocaoglu et al. (2018); Goudet et al. (2018); Pawlowski et al. (2020) and others. Subsequently, this approach has been adopted in various works for computing counterfactuals in applications such as natural language processing (Hu & Li, 2021) and bias reduction (Dash et al., 2022). Other recent works have explored the use of graph neural networks (Sanchez-Martin et al., 2022), normalizing flows (Khemakhem et al., 2021; Javaloy et al., 2023) or diffusion models (Sanchez & Tsaftaris, 2022) to construct structural causal models.

In the present work, we employ variational auto-encoders and normalizing flows to construct deep structural causal models (outlined in § 2.2). Nevertheless, we regard the design choices within our implementation as agnostic to various choices of architecture. Specifically, we deem our approach applicable to any deep structural causal model architecture that yields a reduced-form that is both invertible and differentiable.

