# OpenReview forum: "Deep Backtracking Counterfactuals for Causally Compliant Explanations"
_ICLR.cc/2024/Conference — Submitted to ICLR 2024_

### Official Review · Reviewer_6DZ1 · 2023-10-22

**Soundness:** 3 good
**Presentation:** 4 excellent
**Contribution:** 2 fair
**Rating:** 6
**Confidence:** 4

**Summary:**

This work focuses on *backtracking* counterfactuals as opposed to *interventional* counterfactuals that are most frequently considered. It formulates backtracking counterfactuals as constrained optimization using bijective deep structural causal models. Furthermore, it highlights connections to causally compliant counterfactual explanations.

**Strengths:**

The writing is super clear. I want to thank the authors for such clarity.
I also liked the connection made between backtracking counterfactuals and causally compliant counterfactual explanations.

**Weaknesses:**

The authors make a very big assumption that they only gently brush in section 2.2:
> We assume that all f_i are given as deep generative models.

This assumption is central, and key to feasibility of their formulation. However, structural causal equations are unknown, and often impossible to figure out solely based on data [1].
My question to authors regarding this: `How can we get access to structural causal equations underlying our data? How do we ensure identifiability of such equations given observed data?` I will reconsider my score if authors provide a convincing answer.

Some thoughts regarding this matter: Although authors assume access to underlying generation mechanisms, my guess is that this assumption might not be necessary. identifiability of backtracking counterfactuals seems easier than identifiability of interventional counterfactuals, and we might not need access to generation mechanisms for them. This is just my intuition, and not rigorous.

[1] [Counterfactual Non-identifiability of Learned Structural Causal Models](https://arxiv.org/abs/2301.09031)

**Questions:**

1. How is  *Deep Invertible Structural Causal Models* defined in section 2.2 or structural causal models with invertible reduced form mentioned in the last sentence of Appendix. D different from BGMs [2]? If they are similar, perhaps you can use some of their identifiability results?
2. In the third line of section 2.3., It took me a while to understand the quoted question. I enclosing $x^*$ with parentheses will help the reader.
3. Why does the title of section 3 contain the word `Deep`? I think your formulation as a constrained optimization is not limited to deep neural networks.
4. In the first line of section 3, why do you use the word `example`? I thought counterfactual explanations explained in the former section are just an application of backtracking counterfactuals. If that is the case, I don't think you should use them when explaining your general formulation.
5. I beleive F should be changed to $F^{-1}$ in the end of equation (4).
6. What is $Y$ in the footnote of page 4? I don't think it's mentioned anywhere else in the text.
7. How computationally expensive is calculating the Jacobian esp. for high-dimensional generation mechanisms such as those of images? Can you provide some numbers? I think this is important esp. for practitioners as you may need many Jacobian calculations for your method to converge.
8. How many iterations does it take for Algorithm 1 to converge? Can you provide some ballparks?
9. How did you choose $\lambda=10^4$? How should we choose it for a new domain? What are the implications of large or small $\lambda$s?


[2] [Counterfactual Identifiability of Bijective Causal Models](https://arxiv.org/abs/2302.02228)

---

> ### Author Response · Authors · 2023-11-19
> **Response to Reviewer 6DZ1**
>
> We are delighted to hear that you appreciated the clarity and presentation of the paper and the connection made to the counterfactual explanations literature. We incorporated your valuable comments and suggestions in our revised version.
>
> > The authors make a very big assumption that they only gently brush in section 2.2: We assume that all $f_i$ are given as deep generative models. This assumption is central, and key to feasibility of their formulation. However, structural causal equations are unknown, and often impossible to figure out solely based on data [1]. My question to authors regarding this: How can we get access to structural causal equations underlying our data? How do we ensure identifiability of such equations given observed data? I will reconsider my score if authors provide a convincing answer.
>
> Thank you for pointing this out! We agree that the structural equations are, in general, not identifiable from observational data---even without confounding and with a known causal graph, as assumed in our case. However, recent work [3] has shown that, under the additional assumption of univariate real-valued exogenous variables and diffeomorphic mechanisms $f_i$---both common assumptions in the context of deep generative modelling---the true structural equations can be identified up to an invertible component-wise transformation of the exogenous variables. We included this fact as a footnote into our paper in Section 2.2.
>
> We hope that you find our answer convincing. Please let us know should this not be the case.
>
> [3] Adrian Javaloy, Pablo Sanchez-Martın, and Isabel Valera. Causal normalizing flows: from theory to
> practice. Advances in Neural Information Processing Systems, 2023.
>
> > identifiability of backtracking counterfactuals seems easier than identifiability of interventional counterfactuals, and we might not need access to generation mechanisms for them.
>
> This is an interesting thought. In our setting, we do know the causal graph and the structural equations are identifiable for univariate exogenous variables, as responded above. However, in settings where the causal graph is not known, identifiability does not hold in general [4]. We thank you for pointing this out, and we touch upon this point in the additional paragraph we included in the revised discussion section (Sec. 6).
>
> [4] Amir-Hossein Karimi, Julius Von Kügelgen, Bernhard Schölkopf, and Isabel Valera. Algorithmic recourse under imperfect causal knowledge: a probabilistic approach. Advances in Neural Information Processing Systems, 2020.
>
> Below, we address your remaining questions (where appropriate, we incorporated your suggestions into the revised manuscript).
>
> 1. Indeed, our models fit into the definition of BGMs and we included this fact into our revised manuscript for clarity. It could be the case that backtracking counterfactuals are identifiable for this class as well. We leave this question open for future work.
>
> 2. Incorporated into the revised manuscript.
>
> 3. We would like to stress that this is the formulation that we employ for our approach. It is true, however, that this formulation can also work for shallow methods. In order to stress that the considered formulation is a design choice and not inherent to backtracking in general, we would leave it as it is.
>
> 4. Incorporated into the revised manuscript.
>
> 5. You are correct. Thanks for catching this.
>
> 6. Incorporated into the revised manuscript.
>
> In response to your questions 7. - 8., we included an additional section in Appendix B.1 to elaborate more on these aspects and run additional numerical experiments.

---

> > ### Comment · Reviewer_6DZ1 · 2023-11-21
> > **Reviewer Response**
> >
> > Thanks for your detailed answers.
> > Regarding the identifiability discussion, how do the assumptions in [3] apply to the tasks considered in this work which are mostly focused on images? For example, if my understanding is correct, assumptions in [3] imply that all variables are continuous which doesn't hold for age or gender.

---

> > > ### Author Response · Authors · 2023-11-21
> > >
> > > [3] does generalize to discrete variables (see Section 3.1 of [3]). We, however, do employ continuous variables for CelebA. Rather than working with the binary attributes, we use the continuous logits that we obtain from classifiers that predict the binary attributes from the image (see Section 4.2 in our manuscript). Please let us know should anything remain unclear.

---

> > > > ### Comment · Reviewer_6DZ1 · 2023-11-21
> > > > **Reviewer Response**
> > > >
> > > > Thanks for your answer. I increased my score.
> > > >
> > > > I think it's helpful if you include a discussion about the implications of non-identifiability for your work; i.e., what happens if a practitioner uses this methodology in non-identifiable cases.

---

### Official Review · Reviewer_8kK7 · 2023-10-24

**Soundness:** 2 fair
**Presentation:** 3 good
**Contribution:** 2 fair
**Rating:** 6
**Confidence:** 3

**Summary:**

In this paper, the author proposed a scheme for computing backtracking counterfactuals in SCM.

**Strengths:**

1. The paper is well-written and is easy to follow.

2. The backtracking counterfactual is an interesting concept to explore.

**Weaknesses:**

1. First of all, there should be a related work section in the main paper.  I checked the one in the appendix, and it is still not sufficient. If the contribution of this paper is to introduce a new computation scheme for backtracking counterfactuals, you should at least include how existing work computes it. Also, there is a lack of proper citation in the introduction when you compare your work to the existing methods.

2. My main concern is how significant the proposed method is. It seems like the author optimizes it in the latent space rather than the feature space. Basically, all deep models are learning representations in the latent space, it looks like the proposed algorithm is just an implementation.

3. In the experiment section, the author compares backtracking intervention and conventional intervention, it is more like an introductory article that introduces backtracking intervention.

**Questions:**

See weakness, and also

How does deepBC perform against other algorithms that also conduct backtracking counterfactual reasoning?

---

> ### Author Response · Authors · 2023-11-19
> **Response to Reviewer 8kK7**
>
> We are happy to hear that you find our paper easy to follow and are interested in the backtracking concept. We incorporated your comments and suggestions in our revised version of the manuscript.
>
> We would like to address your points:
>
> > First of all, there should be a related work section in the main paper. I checked the one in the appendix, and it is still not sufficient. If the contribution of this paper is to introduce a new computation scheme for backtracking counterfactuals, you should at least include how existing work computes it. Also, there is a lack of proper citation in the introduction when you compare your work to the existing methods.
>
> Thank you for the suggestion. In response to your point, we decided to move other aspects of our method to the appendix in favor of a compressed related work section (Sec. 5) in the main article. We do keep a more comprehensive related work section in the appendix (App. E), which we also extended. We agree that a comprehensive review of the literature is crucial.
>
> > My main concern is how significant the proposed method is. It seems like the author optimizes it in the latent space rather than the feature space. Basically, all deep models are learning representations in the latent space, it looks like the proposed algorithm is just an implementation.
>
> We believe that you missunderstood the main contribution of our work and are sorry that we did not formulate this more clearly. Our optimization is not just about the latent space of a deep model. The relevant aspect is that this latent space is **embedded into a causal model**, which makes the key difference to these many existing works that you have in mind. This causal model cannot be learned from data without any assumptions, which is why models without the causal models do not achieve the same amount of causal faithfulness as ours. We introduced three metrics in our revised manuscript, one of which measure causal compliance (see Sec. 4.2). Our model performs best here (see Tab. 1), especially in comparison to models **without a causal model**. Please let us know if this is still not clear.
>
> > In the experiment section, the author compares backtracking intervention and conventional intervention, it is more like an introductory article that introduces backtracking intervention.
>
> We agree that this section is also meant to introduce these concepts to readers who are not familiar with causality. To the best of our knowledge, *backtracking interventions* do not exist. However, the experiments also demonstrate several other properties. For example, we show that counterfactuals can be identical for interventional counterfactuals and backtracking counterfactuals, in some settings. In our revised manuscript, the Morpho-MNIST furthermore serves to show the weighting feature of DeepBC (see Sec. 3.3 and App. D.1 in the revised version).
>
> We would furthermore like to address your question:
>
> > How does deepBC perform against other algorithms that also conduct backtracking counterfactual reasoning?
>
> We are not aware of any other method that implements backtracking counterfactuals. As mentioned above, DeepBC is not simply about optimizing in an arbitrary learned latent space. The **causal model** is the essential part. In our revised manuscript, we compare our method to a method without a causal model and quantitatively show that it does not achieve the same degree of causal faithfulness as DeepBC. In our revised manuscript, we introduced three metrics to compare our method against others and found that our method outperforms others in the metric relevant for backtracking (see Tab. 1).

---

> > ### Comment · Reviewer_8kK7 · 2023-11-22
> >
> > Thank you for the clarification and much of my questions have been resolved. Thus I'll increase my score.

---

### Official Review · Reviewer_5aYw · 2023-10-29

**Soundness:** 3 good
**Presentation:** 3 good
**Contribution:** 2 fair
**Rating:** 3
**Confidence:** 2

**Summary:**

The paper introduces a new method to compute the backtracking counterfactuals in structural causal models with deep generative components. The problem is transformed into a tractable optimization in the structured latent space. Through experiments on the data sets MINST and CelebA, the paper also demonstrates that the proposed method has good properties such as being versatile and modular.

**Strengths:**

A good addition to the existing literature.

**Weaknesses:**

1. The importance of the problem under consideration is not well-articulated.
2. The performance evaluations lack a quantitative measure to demonstrate the validity of the method. In other words, how do we know if the generated counterfactuals are good or bad?

**Questions:**

1. Why is it important to generate backtracking counterfactuals?

2. Can the authors provide an important application of their proposed method and demonstrate that the proposed deep backtracking counterfactuals approach provide great solutions for this application?

---

> ### Author Response · Authors · 2023-11-19
> **Response to Reviewer 5aYw**
>
> We are delighted to hear that you consider our work a good addition to the existing literature. We would like to incorporate your valuable comments and suggestions for our revised version. We hope that we will be able to convince you to re-assess the current score.
>
> > The importance of the problem under consideration is not well-articulated.
>
> Thank you for your feedback. Due to the preservation of causal mechanisms, backtracking counterfactuals allow for gaining faithful insights into the structural relationships of the data generating process, which render them a promising opportunity in practical domains such as medical imaging [1], biology [2], and robotics [3]. We incorporated additional text in the introduction (Sec. 1) as a response to your comment.
>
> > The performance evaluations lack a quantitative measure to demonstrate the validity of the method. In other words, how do we know if the generated counterfactuals are good or bad?
>
> This is a relevant point. The short answer is that a general score does not exist. Whether a counterfactual is good or bad highly depends on the application at hand. If we wish to gain faithful insights into **how different variables interact with each other** through the construction of counterfactuals, backtracking presents a favorable alternative over other approaches such as interventional counterfactuals or methods that do not employ a causal model. This is because backtracking is guaranteed to always preserve all causal relationships between variables. To highlight this better, we revised the visualization in the CelebA section (Section 4.2, Fig. 5).
>
> Furthermore, we included a metric to quantitatively assess this property (among two other metrics) and show that our method outperforms the others (Tab. 1).
>
> > Can the authors provide an important application of their proposed method and demonstrate that the proposed deep backtracking counterfactuals approach provide great solutions for this application?
>
> Our method has great potential for applications such as medical imaging, biology and robotics, where domain experts can use our approach to develop and verify insights into their data. For instance, our method could be applied to understand how variables such as age, sex and brain volume interact with a brain MRI image. Rather than creating counterfactuals by breaking links between variables, our method can change the value of brain volume and faithfully take into account how the upstream variable sex changes (which in turn has other downstream effects that are respected) as to preserve the causal mechanism as good as possible. We agree that a real-world application of our method, e.g. in medical imaging, is a very interesting and relevant direction for future work, but beyond the scope of the current work. In our manuscript we use CelebA and Morpho-MNIST as our benchmarks, which makes our work accessible to a broad audience that does not require a biological/medicine background. Our benchmarks are therefore intuitive and interpretable by a broad audience and illustrate the beneficial properties of backtracking counterfactuals.
>
> We would be happy to hear from you whether you are convinced by our reply and the changes to the manuscript. We are happy to make further adjustments.
>
> [1] Cathie Sudlow, John Gallacher, Naomi Allen, Valerie Beral, Paul Burton, John Danesh, Paul
> Downey, Paul Elliott, Jane Green, Martin Landray, et al. Uk biobank: An open access resource for
> identifying the causes of a wide range of complex diseases of middle and old age. PLoS medicine,
> 12(3):e1001779, 2015.
>
> [2] Karren Dai Yang, Anastasiya Belyaeva, Saradha Venkatachalapathy, Karthik Damodaran, Abigail
> Katcoff, Adityanarayanan Radhakrishnan, G. V. Shivashankar, and Caroline Uhler. Multi-domain
> translation between single-cell imaging and sequencing data using autoencoders. Nature Communications, 12(1):31, 2021.
>
> [3] Ossama Ahmed, Frederik Träuble, Anirudh Goyal, Alexander Neitz, Yoshua Bengio, Bernhard
> Schölkopf, Manuel Wüthrich, and Stefan Bauer. Causalworld: A robotic manipulation benchmark for causal structure and transfer learning. International Conference on Learning Representations, 2021.

---

> > ### Comment · Reviewer_5aYw · 2023-11-21
> >
> > Thanks the authors for the detailed response. As the authors commented in the response "Whether a counterfactual is good or bad highly depends on the application at hand", I wonder, given a specific application problem at hand, how could one judge whether we should adopt the method?

---

> > > ### Author Response · Authors · 2023-11-21
> > >
> > > You should employ our method if you would like to create counterfactuals that **retain the causal relationships between all variables**. These could be "how sex and age affect beard", or for example in a medical setting "how age affects brain volume". As we show in our experiments, our method generates counterfactuals that retain those relationships the best for CelebA (Tab. 1 in the revised manuscript).

---

> > > > ### Comment · Reviewer_5aYw · 2023-11-23
> > > >
> > > > But how do we know the counterfactuals created by your method could reliably "retain the causal relationships between all variables"?

---

> > > > > ### Author Response · Authors · 2023-11-23
> > > > >
> > > > > The causal relationship is retained because for our method it always holds by definition for all $i$ that $x_i^* = f_{i}(x^*_{\text{pa}(i)}, u_i^*)$, for a choice of $u_i^*$ that is *close* to the factual realization $u_i$. Thus, the counterfactual world here is where noise $u_i$ is manipulated the least such that antecedent is rendered true, subject to all causal mechanisms $f_i$ being retained.
> > > > >
> > > > > We hope that it is clear now.

---

### Official Review · Reviewer_xQRn · 2023-10-31

**Soundness:** 2 fair
**Presentation:** 4 excellent
**Contribution:** 2 fair
**Rating:** 5
**Confidence:** 4

**Summary:**

The paper proposes a particular instantiation of backtracking counterfactuals introduced by von Kugelgen et al. by formalising the counterfactual generation as a tractable constrained optimisation problem in the latent space of a causal model.

**Strengths:**

- The proposed method draws interesting connections between counterfactuals in explainability literature and causal literature and shows how backtracking counterfactuals can be seen as a generalised form of other.
- The paper is very well written easy to follow the framework introduced
- Nice illustrative examples on the Morpho-MNIST dataset

**Weaknesses:**

- The proposed approach solves an optimisation problem for every counterfactual query, very similar to [1, 2]. Unlike referred papers, the authors here aim to generate faithful counterfactuals respecting the given causal graph; it is unclear how the faithfulness of generated counterfactuals is maintained. (based on results, in the case of celebA it seems like the causal graph is not respected).
- Is identity preservation enforced in the inference optimisation iteration? (as observed in celebA dataset, changing baldness is also affecting gender, facial hair, and age)
- The main difference between the counterfactual explanations and deepBC is that explainability approaches do not use the auxiliary causal variables to generate counterfactual images (at least to the best of my knowledge, I haven't seen any papers using them); using deepBC for generating explanations will severally limit the applicability on datasets with meta information on auxiliary variables and the data generating graph.
- From an explainability point of view, the metrics like faithfulness, reliability, and robustness of the generated counterfactuals would be interesting to discuss, celebA results suggest that these metrics would be affected
- It would be useful to have auxiliary models evaluating the identity preservation and faithfulness of the generated counterfactuals and have a comparison against interventional counterfactuals.


[1] Olah, C., Satyanarayan, A., Johnson, I., Carter, S., Schubert, L., Ye, K. and Mordvintsev, A., 2018. The building blocks of interpretability. _Distill_, _3_(3), p.e10.

[2] Bau, D., Liu, S., Wang, T., Zhu, J.Y. and Torralba, A., 2020. Rewriting a deep generative model. In _Computer Vision–ECCV 2020: 16th European Conference, Glasgow, UK, August 23–28, 2020, Proceedings, Part I 16_ (pp. 351-369). Springer International Publishing.

**Questions:**

Please refer to weakness section

---

> ### Author Response · Authors · 2023-11-19
> **Response to Reviewer xQRn**
>
> We are happy to hear that you appreciate the connection we draw between our method and other methods in the counterfactual explanation literature. We considered your comments and suggestions for our revised version and hope that we will be able to convince you of the relevance of our work.
>
> > The proposed approach solves an optimisation problem for every counterfactual query, very similar to [1, 2]. Unlike referred papers, the authors here aim to generate faithful counterfactuals respecting the given causal graph; it is unclear how the faithfulness of generated counterfactuals is maintained. (based on results, in the case of celebA it seems like the causal graph is not respected).
>
> We regret that the experiments on CelebA do not convey the preservation of the causal laws. In response, we updated Section 4.2 (CelebA experiments). Specifically, the new visualization (Fig. 5) highlights how exactly causal relationships are violated/preserved for the different approaches. In addition, we introduced and evaluated three metrics to compare the methods quantitatively. Our methods performs best on the metric relevant for backtracking counterfactuals. We would be happy to hear from you whether it is clearer now.
>
> > Is identity preservation enforced in the inference optimisation iteration? (as observed in celebA dataset, changing baldness is also affecting gender, facial hair, and age)
>
> Identity preservation is encouraged through the distance function $d(u_i, u_i^*)$. In response to your point, we introduced and assessed the metric "obs" that assesses how much the observed attributes change for the different approaches. Interestingly, our method actually does allow for adjusting the level of preserving the different attributes manually. We did not touch upon this in the current version, but implemented this features in the revised version (see Sec. 3.3 and App. D.1). We hope you will find this an interesting new aspect of our approach.
>
> > The main difference between the counterfactual explanations and deepBC is that explainability approaches do not use the auxiliary causal variables to generate counterfactual images (at least to the best of my knowledge, I haven't seen any papers using them); using deepBC for generating explanations will severally limit the applicability on datasets with meta information on auxiliary variables and the data generating graph.
>
> This is a relevant point that we do not touch upon enough in the current version. There exists a highly active area of research known as *causal representation learning* that is concerned with extracting this meta data from the observed data set. Many conditions for the identifiability of these variables (and causal relationship) have already been established in recent years and we hope that further developments in this field will broaden the applicability of our methods further. To address this point, we decided to include an additional paragraph into our discussion section (Sec. 6).
>
> > From an explainability point of view, the metrics like faithfulness, reliability, and robustness of the generated counterfactuals would be interesting to discuss, celebA results suggest that these metrics would be affected. It would be useful to have auxiliary models evaluating the identity preservation and faithfulness of the generated counterfactuals and have a comparison against interventional counterfactuals.
>
> We agree that we did not discuss the properties of our methods enough in the original manuscript. In response to your comment, we decided to introduce three metrics that measure 1) observable attribute preservation (as mentioned above) and 2) faithfulness with respect to the causal mechanisms and 3) plausibility of the generated counterfactuals. These are the metrics that we are most concerned about in this work and indeed, our approach performs best on 2) (see Tab. 1 in our revised manuscript).

---

> ### Comment · Reviewer_xQRn · 2023-11-23
>
> Thank you very much for your thoughtful rebuttal.
>
> Thanks for highlighting the violated relations; I feel discussing these relations are violated would be really helpful.
> I'm unsure how the introduced metrics will play a role in downstream explanation tasks. Thus, I'll be maintaining my score.

---

### Official Review · Reviewer_GJzV · 2023-11-07

**Soundness:** 3 good
**Presentation:** 3 good
**Contribution:** 2 fair
**Rating:** 5
**Confidence:** 4

**Summary:**

The paper implements backtracking counterfactuals [1] in the framework of deep SCMs [2] and points to connections with counterfactual explanations for deep learning models. The paper performs experiments similar to the MorphoMNIST experiments in [2] as well as showing the differences of interventional and backtracking counterfactuals on CelebA.

[1] Von Kügelgen, Julius, Abdirisak Mohamed, and Sander Beckers. "Backtracking counterfactuals." Conference on Causal Learning and Reasoning. PMLR, 2023.
[2 ]Pawlowski, Nick, Daniel Coelho de Castro, and Ben Glocker. "Deep structural causal models for tractable counterfactual inference." Advances in Neural Information Processing Systems 33 (2020): 857-869.

**Strengths:**

The paper is well written and easy to follow. It combines the idea of backtracking counterfactuals with the deep SCM framework to tackle the problem of counterfactual estimation. The paper is sound and shows convincing results of the differences between interventional and backtracking counterfactuals in high-dimensional settings.

**Weaknesses:**

The paper is a relatively simple combination of [1] and [2] and it is unclear how important the linearisation of the optimisation procedure is for the performance of the counterfactuals: How well would simple SGD perform? Is the optimisation over u* converging or simply stopped after 30 iterations?
From my understanding, the DeepBC algorithm is limited to handle continuous variables and it is unclear how the choice of distance metric would influence the resulting counterfactuals in case of u's with different dimensionalities along the backtracking trace. Additional ablations comparing the different design choices would strengthen the paper.
Additionally, I would encourage the authors to consider adding quantitative evaluations such as proposed by [3].

Generally, it feels like this paper tries to introduce DeepBC as a combination of [1] and [2], while being a counterfactual explanation paper. I am not sure which of the two it really ends up being. Furthermore, the use of the DSCM framework could be highlighted more.

[3] Monteiro, Miguel, et al. "Measuring axiomatic soundness of counterfactual image models." The Eleventh International Conference on Learning Representations. 2022.

**Questions:**

- I saw in the code that the autoencoders use a simple MSE loss without properly modelling the observational noise. Why is this choice made?
- In eq 7 you use an L2 penalty for the distance in latent space. This is very restrictive and would only sensibly work for continuous noise variables.:
  - What's the impact on that if the variables have different noise dimensions?
  - How would this generalise to discrete variables?
  - This does not use any scaling between the variables. In footnote 2, it is mentioned that this isn't necessary because they all assume the same base distribution. I believe this point should be elevated beyond a footnote as it might be lost to readers otherwise.
- You mention that you approximate $F_s$ at $\bar{u}$. Is this correct? What is $\bar{u}$ in this case?
- Why do you use the linearisation? Is this faster than SGD? Did you compare the two?
- Why do we want to enforce sparse changes in $u$? Is this sparse in the dimensions of all u or sparse in the different variables?
- You mention that interventional CF can generate samples in low-density regions and mention it as a weakness. Why is that? Isn't that actually one of the strengths in terms of disentanglement and generalisation?
- You mention that you're using standardized logits for modelling. How do you compute them? What's the impact of using this over discrete variables?
- You mention the composability / modularity as a strength of your work. However, this is generally possible within any causal generative framework, particularly the DSCM framework. What's special about this work in this regard?
- You mention you're using "an image regressor, together with an unconditional AE ...". Can you elaborate how this baseline works?
- You mention that DeepBC does not properly sample from the counterfactual distribution and it didn't yield satisfactory results. Why is that? What were the results?
- As for CF explanations, how does this compare to explaining anti-causal predictions by using causal generative models (e.g. see [4]).

[4] Zhang, Cheng, Kun Zhang, and Yingzhen Li. "A causal view on robustness of neural networks." Advances in Neural Information Processing Systems 33 (2020): 289-301.

---

> ### Author Response · Authors · 2023-11-19
> **Response to Reviewer GJzV - Part I**
>
> We are delighted to hear that you found our work to be well-written and easy to follow. Furthermore, we appreciate the effort you undertook in rigorously assessing our manuscript. We would like to consider your valuable comments and suggestions for our revised version and hope that we will be able to convince you of the relevance of our work.
>
> > The paper is a relatively simple combination of [1] and [2]
>
> The main contribution of our work is to provide a practical and computationally efficient way of generating backtracking counterfactuals. It is important to note that the formulation presented in [2] cannot be readily implemented “out-of-the box”, since it requires computing the posterior $p(\mathbf{u}^* | \mathbf{u}, \mathbf{x}^*_S)$ (see App. A.2), which involves several steps that are computationally intractable. Our approach computes the mode of the posterior $p(\mathbf{u}^* | \mathbf{u}, \mathbf{x}^*_S)$, and shows that this can, in fact, be done by solving a constrained optimization problem that is tractable (under the assumption of (approximate) bijectivity of the generative models). Our research goes therefore well beyond the results of [2] due to its focus on numerical evaluation, computation, and practical evaluation on Morpho-MNIST and CelebA.
>
> Our approach indeed relies on incorporating deep neural networks into structural causal models, but only as a starting point (this is not the contribution of the paper). The work therefore builds on new advances, such as [1], which makes our work timely and relevant to the community.
>
> > it is unclear how important the linearisation of the optimisation procedure is for the performance of the counterfactuals: How well would simple SGD perform? Is the optimisation over u* converging or simply stopped after 30 iterations? From my understanding, the DeepBC algorithm is limited to handle continuous variables and it is unclear how the choice of distance metric would influence the resulting counterfactuals in case of u's with different dimensionalities along the backtracking trace. Additional ablations comparing the different design choices would strengthen the paper.
>
> Thank you for this remark. We agree that we only swiftly touched upon this part in the old manuscript. We confirm that the optimization indeed converges after 30 iterations with respect to the penalty loss (equation (7)). Empirically, we observe that our method converges much quicker, compared to gradient-based optimization algorithms. Note that, strictly speaking, SGD is not applicable, because the optimization is not over a data set, but only a single example. To address your point, we included an additional section to our appendix (App. B.1) where we describe the choice of algorithm and parameters in detail. We also provide numerical experiments that contrast constraint linearization to the Adam optimizer. These experiments highlight the fact that the convergence of Adam is very sensitive to the step-size, whereas our method works out-of-the-box.
>
> > it is unclear how the choice of distance metric would influence the resulting counterfactuals in case of u's with different dimensionalities along the backtracking trace.
>
> While we agree that choosing the distance metric might be non-trivial in more complex examples such as multi-modal single cell data [4], we in fact, provide an intuitive interpretation of the distance metric as the log of the backtracking conditional density $p(\mathbf{u}^*|\mathbf{u})$ (see App. A2). As a result, the distance metric arises indirectly from the definition of the backtracking counterfactuals.
>
> [4] Karren Dai Yang, Anastasiya Belyaeva, Saradha Venkatachalapathy, Karthik Damodaran, Abigail
> Katcoff, Adityanarayanan Radhakrishnan, G. V. Shivashankar, and Caroline Uhler. Multi-domain
> translation between single-cell imaging and sequencing data using autoencoders. Nature Com-
> munications, 12(1):31, 2021

---

> ### Author Response · Authors · 2023-11-19
> **Response to Reviewer GJzV - Part II**
>
> > Additionally, I would encourage the authors to consider adding quantitative evaluations such as proposed by [3]
>
> We agree that quantitative evaluations are of great interest to the reader. However, the work you mention is based on an axiomatic characterization of interventional counterfactuals (section 7.3.1 in [5]) and it is not obvious to us whether and how it can be applied/adapted to the backtracking philosophy, which is inherently different. Specifically, the reversibility axiom is not obvious to apply to backtracking, because backtracking does not employ functional re-assignmnents, which this axiom is based on (instead, all causal mechanisms are kept in place). In order to address your point, we introduced three metrics (Section 4.2) that measure how well the original example is preserved in terms of observable attributes (obs), plausibility, and causal compliance (causal). We indeed find that our method outperforms other approaches in the causal compliance metric, which is the relevant metric for backtracking counterfactuals (see Tab. 1).
>
> > Generally, it feels like this paper tries to introduce DeepBC as a combination of [1] and [2], while being a counterfactual explanation paper. I am not sure which of the two it really ends up being.
>
> The main part of this point has already been addressed above. We note that the relation of our work to multiple other lines of work is actually a strength, since it is relevant for the causal community, by focusing a high-dimensional data (images) and computational tractability, and it is relevant to the high-dimensional explanations community by providing a solid theoretical foundation rooted in causality (see App. A2). These fields have thus far evolved separately (for the most part), and our work bridges the gap.
>
> > Furthermore, the use of the DSCM framework could be highlighted more.
>
> We modified the first mentioning of deep learning for modelling structural equations to highlight this more. Thank you for the suggestion.
>
> We would furthermore like to address your questions:
>
> >1. I saw in the code that the autoencoders use a simple MSE loss without properly modelling the observational noise. Why is this choice made?
>
> During training, there is no difference compared to the standard variational autoencoder [6] (the fixed observational noise leads to the MSE loss of the decoder.). For computing counterfactuals, we only keep the encoder and decoder of the trained models and do not consider observational noise. This is justified to some extent by [7], as written in the paper. Please let us know if we did not understand your question correctly.
>
> > In eq 7 you use an L2 penalty for the distance in latent space. This is very restrictive and would only sensibly work for continuous noise variables.: What's the impact on that if the variables have different noise dimensions? How would this generalise to discrete variables? This does not use any scaling between the variables. In footnote 2, it is mentioned that this isn't necessary because they all assume the same base distribution. I believe this point should be elevated beyond a footnote as it might be lost to readers otherwise.
>
> In our method, we employ generative models only, which all operate on a standard Gaussian as base distribution. In response to your comment, we decided to include this into the main text (see Sec. 3.3). It is true that the method does not readily support discrete variables. We note, however, that discrete variables can often be replaced by logits , as we do in the experiments for CelebA (see Sec. 4.2.: Experimental Setup). In this case, our method can be applied.
>
> Our framework also allows for scaling the distances for variables differently. We find this an excellent point and therefore included this idea in the revised version of the manuscript (see equ. (8) and experiments in App. D1).
>
> [5] Judea Pearl. Causality. Cambridge University Press, 2009.
>
> [6] Diederik P. Kingma and Max Welling. Auto-Encoding Variational Bayes. International Conference on Learning Representations, 2014.
>
> [7] Patrik Reizinger, Luigi Gresele, Jack Brady, Julius von Kügelgen, Dominik Zietlow, Bernhard Schölkopf, Georg Martius, Wieland Brendel, and Michel Besserve. Embrace the Gap: VAEs
> Perform Independent Mechanism Analysis. Advances in Neural Information Processing Systems,
> 35:12040–12057, 2022.

---

> > ### Author Response · Authors · 2023-11-19
> > **Response to Reviewer GJzV - Part III**
> >
> > > You mention that you approximate $F_S$ at $\bar{u}$. Is this correct? What is $\bar{u}$ in this case?
> >
> > You are correct, and we agree that this may not have been clear in the original manuscript. In optimization, it is common to construct an update rule that is based on the previous update. $\bar{\mathbf{u}}$ is always the previous update, starting with $\bar{\mathbf{u}} = \mathbf{u}_0$. We made this point clearer in the updated version of the text (see also Alg.1).
> >
> > > Why do you use the linearisation? Is this faster than SGD? Did you compare the two?
> >
> > Linearization converges indeed faster than first-order methods, in our experiments. In response to your question, we included a section to the appendix where we elaborate on this and show numerical experiments (App. B.1).
> >
> > > Why do we want to enforce sparse changes in $u$? Is this sparse in the dimensions of all $u$ or sparse in the different variables?
> >
> > Regarding your first question: When we wish to obtain insights into the data-generating process (which is what our method is aimed at), it may be desirable to encourage that not all visible variables $\mathbf{x}$ change for interpretability reasons. We hope that this is more clear in our updated visualization for CelebA (Fig. 5).
> >
> > To answer your second question, there could be multiple ways. However, we generally consider all $u$ that belong to one observable $x$. In this sense, your second option (sparsity in the different variables) is right.
> >
> > > You mention that interventional CF can generate samples in low-density regions and mention it as a weakness. Why is that? Isn't that actually one of the strengths in terms of disentanglement and generalisation?
> >
> > We believe that you misunderstood this point and we are sorry that this wasn't laid out precise enough. We do not intend to claim that backtracking counterfactuals are generally better than interventional counterfactuals. We believe that both approaches can give valuable insights. The preference should be established based on the specific use case. In algorithmic recourse [8], for instance, interventional counterfactuals are of greater interest, as written in our related work section. Backtracking may be more favorable for applications of developing insights into the structural relations between different variables. This can trivially not be achieved by interventional counterfactuals, since they break the governing laws. We updated the phrasing on our manuscript (Sec. 4.1) to make this point clearer.
> >
> > > You mention that you're using standardized logits for modelling. How do you compute them? What's the impact of using this over discrete variables?
> >
> > We generate the logits via classifiers that predict the variables as functions of the image pixels. This is our approach for handling non-continuous variables. The consequence is that we never receive binary counterfactuals. Instead, we always obtain counterfactual logits, which can be translated back into probabilities. I.e., we obtain the probability of the person to be male in the counterfactual rather than the discrete variable directly.
> >
> > > You mention the composability / modularity as a strength of your work. However, this is generally possible within any causal generative framework, particularly the DSCM framework. What's special about this work in this regard?
> >
> > This is a strength in comparison to counterfactual explanation methods that do not operate on the DeepSCM framework, which is why we elaborate on this point in Section 3.2. We explicitly highlight this as a strength *in the context of counterfactual explanations*.
> >
> > > You mention you're using "an image regressor, together with an unconditional AE ...". Can you elaborate how this baseline works?
> >
> > This baseline is adapted from how standard counterfactual explanation methods work. These methods do not employ a causal model. Rather, these methods are aimed at minimizing some distance function between $\mathbf{x}$ and $\mathbf{x}^*$ such that an image regressor yields a specific, but different prediction than $\mathbf{x}$. To this end, we employ a VAE that is not embedded into a causal model. Instead, we only minimize a distance between the latent VAE embeddings of $\mathbf{x}$ and $\mathbf{x}^*$, without the causal embedding.
> >
> > > You mention that DeepBC does not properly sample from the counterfactual distribution and it didn't yield satisfactory results. Why is that? What were the results?
> >
> > Our method only considers the mode of the underlying backtracking posterior distribution $p(\mathbf{u}^* | \mathbf{u}, \mathbf{x}^*_S)$ (see App.A2). The backtracking-posterior is computationally intractable.
> >
> > [8] Amir-Hossein Karimi, Bernhard Schölkopf, and Isabel Valera. Algorithmic Recourse: from Coun-
> > terfactual Explanations to Interventions. In ACM Conference on Fairness, Accountability, and
> > Transparency, pp. 353–362, 2021.

---

> > > ### Author Response · Authors · 2023-11-19
> > > **Response to Reviewer GJzV - Part IV**
> > >
> > > > As for CF explanations, how does this compare to explaining anti-causal predictions by using causal generative models (e.g. see [4]).
> > >
> > > Our method can handle any causal relationship (including anti-causal ones) and can thus be seen as more general. The main key to understanding the relationship between our method and CF explanations is the idea that a predictive machine learning model can be viewed as a structural equation. In this sense, a counterfactual explanation corresponds backtracking with the goal of obtaining insights into this particular structural equation.

---

### Author Response · Authors · 2023-11-19
**General Response**

We thank all reviewers for their time and effort in reviewing our manuscript. All reviewers found our manuscript well-written---"The writing is super clear. I want to thank the authors for such clarity" (`6DZ1`)---and interesting, with reviewers
stating that it constitutes "a good addition to the existing literature" (`5aYw`) with "nice illustrative examples on the Morpho-MNIST dataset"(`xQRn`). We highly value the constructive feedback provided by all reviewers, and incorporated their suggestions in the revised manuscript to further improve our work.

To reiterate the main contribution of our work, we introduce **a new, practical** method for backtracking in deep structural causal models (DeepBC).

We build on prior work that formalized backtracking counterfactuals through **distributions that are generally intractable to evaluate or compute** (see the integrals involved in the 3-step backtracking procedure  described in Appendix A.1).

To address this challenge, DeepBC solves a **tractable optimization problem** involving the latent variables of a deep SCM to **approximately but efficiently generate maximum a posteriori backtracking counterfactuals**.
As a causally grounded explanation method applicable to high-dimensional data, DeepBC fills a gap in the existing literature between **non-causal** explanation tools built for complex data such as images and **causal** methods that have only been applied to simple, low-dimensional settings.

We believe that our method can be of great use in practice by providing an analysis tool for asking counterfactual questions across a wider range of settings. These questions can yield valuable insights into the structural relationships within the data. Possible application domains include medical imaging [1], biology [2] and robotics [3].

We took all the reviewers' comments into account and made the following main changes (highlighted in blue in the revised manuscript):

1. We introduced three metrics to quantitatively compare DeepBC to other methods on CelebA: one score that measures the distance of the factual to the counterfactual realization in terms of *attribute realizations*, one score that measures the distance of the factual to the counterfactual realization in terms of the *latent variables*, and one score that measures the likelihood of the counterfactual under the prior. **DeepBC outperforms all other methods on the second score**, which is the relevant metric for evaluating backtracking counterfactuals.

2. In Section 4.2 (CelebA experiments) we now better highlight the properties of DeepBC compared to other methods. Specifically, we improved the visualization (Fig. 5) to highlight exactly which causal mechanisms are violated by other methods and how this is not the case for DeepBC.

3. In the introduction, we now highlight the practical relevance of preserving causal mechanisms in counterfactual explanations.

4. We  incorporated an additional property of DeepBC, for which we also ran experiments. This feature allows choosing weights for the different variables, which in turn allows for different degrees of preservation for the different variables. These weights can be specified manually.

5. In the new Appendix B.1, we now discuss the choice of the optimization algorithm and the corresponding hyperparameters in greater detail. We also ran numerical experiments to empirically underline the benefits of the employed constraint linearization technique (similar to the Levenberg–Marquardt method) compared to the Adam optimizer.

6. In the discussion section (Sec. 6), we now address the identifiability of the reduced-form and structural equations of the causal model.

7. We included a compressed version of the related work section in the main article (Sec. 5). (The more comprehensive version is kept in the appendix (App. E))

We thus believe to have adequately addressed  all comments by the reviewers. Please let us know should anything remain unclear or if you have further suggestions for improvements.

[1] Cathie Sudlow, John Gallacher, Naomi Allen, Valerie Beral, Paul Burton, John Danesh, Paul Downey, Paul Elliott, Jane Green, Martin Landray, et al. Uk biobank: An open access resource for identifying the causes of a wide range of complex diseases of middle and old age. PLoS medicine, 12(3):e1001779, 2015.

[2] Karren Dai Yang, Anastasiya Belyaeva, Saradha Venkatachalapathy, Karthik Damodaran, Abigail Katcoff, Adityanarayanan Radhakrishnan, G. V. Shivashankar, and Caroline Uhler. Multi-domain translation between single-cell imaging and sequencing data using autoencoders. Nature Communications, 12(1):31, 2021.

[3] Ossama Ahmed, Frederik Träuble, Anirudh Goyal, Alexander Neitz, Yoshua Bengio, Bernhard Schölkopf, Manuel Wüthrich, and Stefan Bauer. Causalworld: A robotic manipulation benchmark for causal structure and transfer learning. International Conference on Learning Representations, 2021.

---

> ### Author Response · Authors · 2023-11-22
> **Request for Reconsideration and Feedback on Revised Manuscript**
>
> We again would like to express our sincere gratitude to all reviewers for the valuable feedback on our manuscript.
>
> As we approach the final deadline, we kindly request all reviewers to reconsider the manuscript in light of the implemented changes, in case they have not done so yet. We believe these revisions effectively address the concerns raised during the initial review process, and we hope that this can be reflected in the final rating.

---

### Meta-Review · Area_Chair_yNKa · 2023-12-11

**Metareview:**

The paper describes a practical algorithm for computing backtracking counterfactuals for deep structural causal models. It is argued that the results preserve important properties such as retaining causal relationships. It is not so easy to compare the proposed approach with all the possible ideas in the literature and the experimental results are somewhat limited to highlight all that the approach can (or not) do. It is a reasonable paper but which landed no great excitement among the committee members within the short reviewing process span.

**Justification For Why Not Higher Score:**

Under the assumption that the committee members are a sample of the audience (with the known risk that this might not be true), the paper could have done a stronger case for its publication, or it is not the best presentation for the venue.

**Justification For Why Not Lower Score:**

N/A

---

### Decision · Program_Chairs · 2024-01-16

Reject